# RNAPII-dependent ATM signaling at collisions with replication forks

Elias Einig [1], Chao Jin [1], Valentina Andrioletti[1,3], Boris Macek [2] & Nikita Popov [1] ✉

Deregulation of RNA Polymerase II (RNAPII) by oncogenic signaling leads to collisions of RNAPII with DNA synthesis machinery (transcription-replication conflicts, TRCs). TRCs can result in DNA damage and are thought to underlie genomic instability in tumor cells. Here we provide evidence that elongating RNAPII nucleates activation of the ATM kinase at TRCs to stimulate DNA repair. We show the ATPase WRNIP1 associates with RNAPII and limits ATM activation during unperturbed cell cycle. WRNIP1 binding to elongating RNAPII requires catalytic activity of the ubiquitin ligase HUWE1. Mutation of HUWE1 induces TRCs, promotes WRNIP1 dissociation from RNAPII and binding to the replisome, stimulating ATM recruitment and activation at RNAPII. TRCs and translocation of WRNIP1 are rapidly induced in response to hydroxyurea treatment to activate ATM and facilitate subsequent DNA repair. We propose that TRCs can provide a controlled mechanism for stalling of replication forks and ATM activation, instrumental in cellular response to replicative stress.

RNAPII drives the expression of protein-coding genes and thereby is essential for all aspects of cell biology. Beyond the functions of individual genes, the genome-wide association of RNAPII with chromatin and several associated factors can differentially impact on genome stability. On one hand, RNAPII is an essential element of the transcription-coupled repair system and can also promote DNA repair, for example via recruitment of homologous recombination-dependent DNA repair factors or production of non-coding RNAs at sites of DNA double-strand breaks[1,2].

On the other hand, the deregulation of RNAPII, for example by oncogenic signaling, interferes with DNA replication, leading to DNA damage and genomic instability. Pervasive stimulation of RNAPII by oncogenes can lead to unbalanced unwinding of DNA, supercoiling and excessive torsional stress[3,4]. Deregulation of RNAPII can also lead to excessive formation of DNA-RNA hybrids (R-loops), which present a major roadblock for replication machinery and considered a key factor of genome instability[5,6]. Both torsional stress and R-loops increase the frequency of collisions between RNAPII and replication machinery (transcription replication conflicts, TRCs). Although unavoidable (for example, for very long genes) and likely readily resolved during the normal cell cycle, increased incidence of TRCs may become a major source of DNA damage upon aberrant activation of RNAPII[7]. For example, oncogenes that deregulate RNAPII and cause TRCs - such as RAS and MYC - also induce replicative stress and DNA damage[8-12]. Depending on the relative orientation of RNAPII and replication fork, TRCs can lead to the activation of distinct DNA damage response kinases−ATR or ATM[13,14].

Different mechanisms that involve RNAPII are instrumental in the resolution of TRCs. On one hand, RNAPII can be evicted to allow replication fork progression[15-17]. On the other hand, RNAPII can promote the recruitment of factors that facilitate the resolution of R-loops, including DNA helicases and spliceosomal RNA helicases, the exonuclease MRE11, DNA repair proteins BRCA1/2 and elongation factor PAF1c[18,19]. Some of these factors are recruited by RNAPII dependent on its functional state and posttranslational modifications. For example, PAF1c and the spliceosome associate with the elongating pool of RNAPII phosphorylated at Ser2 of its carboxy-terminal domain (CTD) by CDK9 and CDK12[20]. The PAF1 complex regulates processive elongation but also promotes the resolution of TRCs and DNA repair[16,21,22]. The spliceosome is instrumental in the resolution of

[1]Department of Medical Oncology and Pulmonology, University Hospital Tübingen, Otfried-Mueller-Str 14, 72076 Tübingen, Germany. [2]Interfaculty Institute of Cell Biology, Eberhard Karls University of Tübingen, Auf d. Morgenstelle 15, 72076 Tübingen, Germany. [3]Present address: enGenome S.R.L., Via Fratelli Cuzio 42, 27100 Pavia, Italy. ✉e-mail: nikita.popov@med.uni-tuebingen.de

R-loops and this function is stimulated by the DNA damage response signaling[19]. These observations suggest that RNAPII phosphorylation can impact the outcome of collisions with the replisome.

A critical pathway controlling RNAPII function is ubiquitination—multiple ubiquitin ligases regulate transcription via modification of RNAPII or regulatory proteins, including general and site-specific transcription factors[23]. The HUWE1 ubiquitin ligase controls RNAPII elongation via several transcription factors, including MYC and β-catenin, which are potent regulators of transcriptional pausing[24–26]. At the same time, HUWE1 was implicated in controlling cellular response to DNA damage and DNA replication stress via regulation of histones H2AX, H2B, and H1, DNA polymerase sliding clamp PCNA and replication initiation factor CDC6[27–29], indicating that it may coordinate DNA replication with RNAPII activity.

Using a genetic system with catalytic inactivation of HUWE1, we show that HUWE1 activity is critical for the resolution of TRCs and suppressing ATM activation. Mechanistically, HUWE1 promotes the association of the ATPase WRNIP1 with the elongating pool of RNAPII, limiting the recruitment of ATM to RNAPII-associated MRN complex. Mutation of HUWE1 leads to dissociation of WRNIP1 from RNAPII, leading to RNAPII-dependent activation of ATM. WRNIP1-RNAPII association is disrupted upon drug-induced replicative stress, inducing TRCs, RNAPII-dependent activation of ATM and repair of replication-associated DNA damage.

## Results

### Catalytic mutation of HUWE1 induces TRCs

To study the role of catalytic activity of HUWE1, we used CRISPR to replace the catalytic cysteine of the HECT domain with either Ser (HUWE1-CS) or Cys as a control (HUWE1-WT) (Fig. 1a; Supplementary Fig. 1a[30]). We established several cell lines and confirmed the correct gene targeting by PCR and Sanger sequencing (Supplementary Fig. 1b). HUWE1 mutation strongly altered gene expression with more than 1900 up- and down-regulated genes, in contrast to control cells

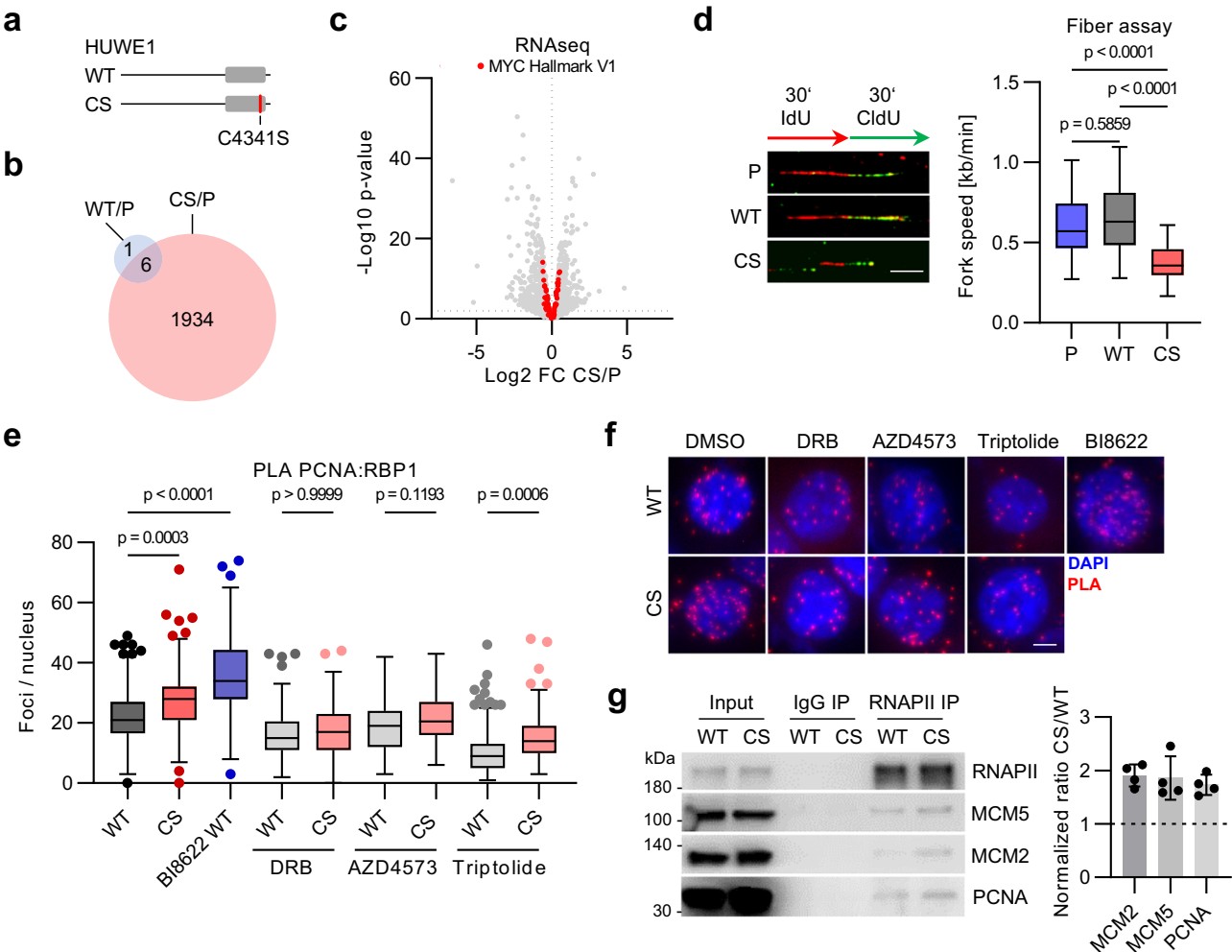

**Fig. 1 | Mutation of HUWE1 induces transcription-replication conflicts. a** The catalytic cysteine of the endogenous HUWE1 HECT domain (gray) was replaced by a serine (HUWE1-CS, CS) or the wild type cysteine (HUWE1-WT, WT) in parental HCT116 cells (P). **b** Transcriptome analysis of HUWE1-P, HUWE1-WT and HUWE1-CS cells. Overlap of genes significantly deregulated (p-value ≤ 0.01, FDR ≤ 0.01, n = 3) in HUWE1-WT or HUWE1-CS cells relative to parental HUWE1-P cells. **c** Volcano plot of RNAseq analysis (n = 11350 transcripts over 3 biological replicates). Members of the MYC hallmark V1 gene set are highlighted in red. **d** DNA fiber assay in parental HCT116 (P), HUWE1-WT and HUWE1-CS cells. Representative images of single DNA fibers (left) and quantification (right) are shown (n = 50 forks per group). Scale bar: 4 μm; n = 3 experiments **e, f** Proximity ligation assay (PLA) with antibodies to PCNA and RNAPII in HUWE1-WT and HUWE1-CS cells treated with 100 μM DRB, 10 nM AZD4573, 100 nM triptolide or the HUWE1 inhibitor BI-8622 (5 μM) for 7 h. **e** Quantification of proximity pairs within nuclei for n = 181,173,154,165,169,147,142,196,152 cells. **f** Representative images. Scale bar: 5 μm. **g** Immunoprecipitation (IP) of RNAPII in formaldehyde-crosslinked and sonicated HUWE1-WT and HUWE1-CS cells. A representative image (left) and quantification of the CS/WT ratio (right) are shown (n = 4, mean ± SD). **d, e** Boxplots show median±quartiles with whiskers ranging up to 1.5-fold of the inter-quartile range. P-values were determined using Kruskal–Wallis test followed by Dunn's multiple comparison. Source data are provided as a Source Data file.

(Fig. 1b, c; Supplementary Fig. 1c). Although MYC was previously characterized as a key HUWE1 target, known MYC-associated genes were not strongly enriched among HUWE1-dependent transcripts (Fig. 1c, Supplementary Fig. 1d, e). Also, depletion of MYC in HCT116 cells deregulated a distinct set of genes compared to HUWE1 mutation (Supplementary Fig. 1e–g), indicating that HUWE1 can control RNAPII function via other effectors. In agreement with previous studies[28], mutation of HUWE1 slowed progression of DNA replication forks, as judged by the DNA fiber assays (Fig. 1d).

We next compared the incidence of transcription-replication conflicts (TRCs) in HUWE1-WT and HUWE1-CS cells using a proximity ligation assay (PLA) with antibodies against PCNA and RNAPII[13]. The PLA signal increased in HUWE1-CS cells compared with control HUWE1-WT cells (Fig. 1e, f). To corroborate this result, we immunoprecipitated RNAPII from lysates of formaldehyde-crosslinked cells, which were briefly sonicated to preserve association of protein complexes proximally positioned on chromatin. Immunoblotting of these precipitates revealed increased levels of replisome proteins MCM2, MCM5 and PCNA in HUWE1-CS cells compared with HUWE1-WT cells (Fig. 1g). Likewise, immunoprecipitation of PCNA and CDC45 yielded increased levels of RNAPII in HUWE1-CS cells compared to HUWE1-WT cells (Supplementary Fig. 1h, i). Chemical inhibition of HUWE1 with BI-8622[25] also potently induced TRCs (Fig. 1e, f). Collisions, induced by mutation of HUWE1 were dependent on transcriptional elongation, since inhibition of the CTD kinase CDK9 with 5,6-dichloro-1-beta-D-ribofuranosylbenzimidazole (DRB) or AZD4573 or inactivation of RNAPII using triptolide[31–33] rescued the increase in the PLA signal (Fig. 1e, f).

## WRNIP1 is a HUWE1 interactor that prevents TRCs

To identify candidate effectors of HUWE1 in suppressing RNAPII-dependent TRCs, we immunoprecipitated endogenous HUWE1 from HUWE1-WT and HUWE1-CS cells and analyzed interacting proteins by LC-MS/MS (Fig. 2a). This experiment identified several known HUWE1 binding partners, including TP53 and enzymes of the α-ketoglutarate dehydrogenase complex[34]. Among the top ten hits was the ATPase WRNIP1 that was shown to bind and stabilize replication forks upon inhibition of DNA synthesis[35–37]. WRNIP1 was further identified in independent, quantitative AP-MS experiments using tandem mass tag (TMT) labeling (Supplementary Fig. 2a). Subsequent immunoprecipitation assays showed a robust interaction of endogenous HUWE1 and WRNIP1 in HCT116 and several other cell lines (Fig. 2b; Supplementary Fig. 2b). Catalytic activity of HUWE1 was not required for HUWE1-WRNIP1 binding and did not affect WRNIP1 turnover rate (Fig. 2b; Supplementary Fig. 2c).

Immunofluorescence experiments showed that HUWE1 and WRNIP1 co-localized in the nucleus (Fig. 2c). PLA assays showed that HUWE1 and WRNIP1 interaction was potentiated in the S phase of the cell cycle, as judged by concomitant pulse labeling with EdU and click reaction with Cy3-azide (Fig. 2d; Supplementary Fig. 2d). To identify regions in WRNIP1 that mediate HUWE1 binding we immunoprecipitated HUWE1 from HCT116 cells, expressing WRNIP1 variants, encompassing different functional domains. This experiment identified the leucine zipper domain of WRNIP1 as the region required for HUWE1 binding (Fig. 2e; Supplementary Fig. 2e, f).

Like HUWE1 mutation, depletion of WRNIP1 reduced DNA replication fork progression, albeit to a smaller extent, indicative of mild replicative stress in shWRNIP1 cells (Fig. 2f). Consistently, WRNIP1 knockdown diminished EdU incorporation in HUWE1-WT cells (Fig. 2g, h; Supplementary Fig. 2g). Depletion of WRNIP1 in HUWE1-CS cells did not further reduce EdU incorporation, which was strongly reduced compared to HUWE1-WT cells (Fig. 2g, h). Depletion of WRNIP1-strongly increased the level of transcription-dependent TRCs, comparable to the effect of HUWE1 mutation (Fig. 2i). The WRNIP1 variant lacking the LZ domain that is required for HUWE1 binding also induced conflicts, showing that interaction with HUWE1 is important for TRC

resolution (Fig. 2j). Unexpectedly, depletion of WRNIP1 in HUWE1-CS cells rescued the increase in TRCs (Fig. 2k). Since both HUWE1 and WRNIP1 were shown to protect replication forks upon stress[28,36], this result indicated that HUWE1 and WRNIP1 may redundantly stabilize forks at sites of collisions. Consistent with this idea, PLA and immunoprecipitation assays showed an increased association of WRNIP1 with MCM2, CDC45, and in HUWE1-CS cells, suggesting that mutation of HUWE1 promotes WRNIP1 binding to replication forks (Supplementary Fig. 2h, i). In turn, depletion of WRNIP1 induced a small increase in HUWE1 association with PCNA (Supplementary Fig. 2j).

## HUWE1 controls association of WRNIP1 with elongating RNAPII

ChIP-seq and Cut&Run experiments with WRNIP1 antibodies revealed that WRNIP1 binds chromatin in a HUWE1-dependent manner (Fig. 3a, b, Supplementary Fig. 3a, b). Consistently, immunoprecipitation of WRNIP1 from benzonase-treated lysates and PLA experiments showed that WRNIP1 associates with RNAPII (Fig. 3c). Mutation of HUWE1 strongly compromised WRNIP1 association with total and pS2-RNAPII but not with the pS5-phosphorylated RNAPII (Fig. 3d), indicating that HUWE1 controls recruitment of WRNIP1 to elongation-competent RNAPII. To show that WRNIP1 binds chromatin in a complex with RNAPII, we performed sequential ChIP assays with WRNIP1 antibodies followed by elution of antibody-bound complexes and immunoprecipitation with RNAPII antibodies. This experiment showed a strong reduction in occupancy at several tested promoter sites in HUWE1-CS cells compared to HUWE1-WT cells (Fig. 3e). Furthermore, sequential immunoprecipitation with HUWE1 antibodies followed by pS2-RNAPII antibodies detected WRNIP1, suggesting that HUWE1 and WRNIP1 simultaneously associate with pS2-RNAPII, most likely as a complex (Supplementary Fig. 3c). WRNIP1 association with RNAPII strictly required the Ub-binding domain and was reduced by the deletion of the LZ domain, which mediates WRNIP1-HUWE1 binding (Supplementary Fig. 3d).

Mutation of HUWE1 and depletion of WRNIP1 increased pS2-RNAPII levels as determined by immunoblotting of whole cell lysates (Supplementary Fig. 3e). Depletion of WRNIP1 in HUWE1-CS cells abolished the increase in S2 phosphorylation, correlating with the effect on TRCs. Since TRCs are dependent on transcription elongation, we reasoned that the increase in pS2-RNAPII may reflect a pool of elongating RNAPII locked at the sites of collisions. In line with this view, immunoprecipitation assays showed preferential association of replisome proteins with pS2-RNAPII compared with pS5-RNAPII in HUWE1-CS cells (Fig. 3f). In contrast, replisome proteins were absent in sequential immunoprecipitations with WRNIP1 antibodies followed by pS2-RNAPII antibodies (Supplementary Fig. 3f). Since straight RNAPII immunoprecipitation readily detects replisome proteins in HUWE1-CS cells (Fig. 1g), this result suggests that WRNIP1-associated RNAPII complexes do not localize at TRCs.

ChIP-seq experiments showed that pS2-RNAPII distribution correlated with WRNIP1 binding (Supplementary Fig. 3g). Depletion of WRNIP1 or mutation of HUWE1 caused accumulation of pS2-RNAPII at 3' regions of the gene bodies and at transcription end sites, which was reverted in the double mutant cells (Fig. 3g, Supplementary Fig. 3h). This pattern of pS2-RNAPII accumulation correlated with the incidence of TRCs in the four cell lines, raising the possibility that TRCs may preferentially occur at the sites. A group of 2221 genes, for which the reciprocal regulation in the single and double mutant cells was most pronounced, showed a stronger HUWE1-dependent recruitment of WRNIP1 than most of the genes (Supplementary Fig. 3i, j). Together, these data indicate that HUWE1-dependent recruitment of WRNIP1 to RNAPII may prevent accumulation of pS2-RNAPII in the gene bodies and transcription-end sites and promote the resolution of TRCs.

## HUWE1 and WRNIP1 suppress ATM signaling

In agreement with activation of ATM signaling by TRCs[13], mutation of HUWE1 induced phosphorylation of ATM and ATM targets KAP1 and

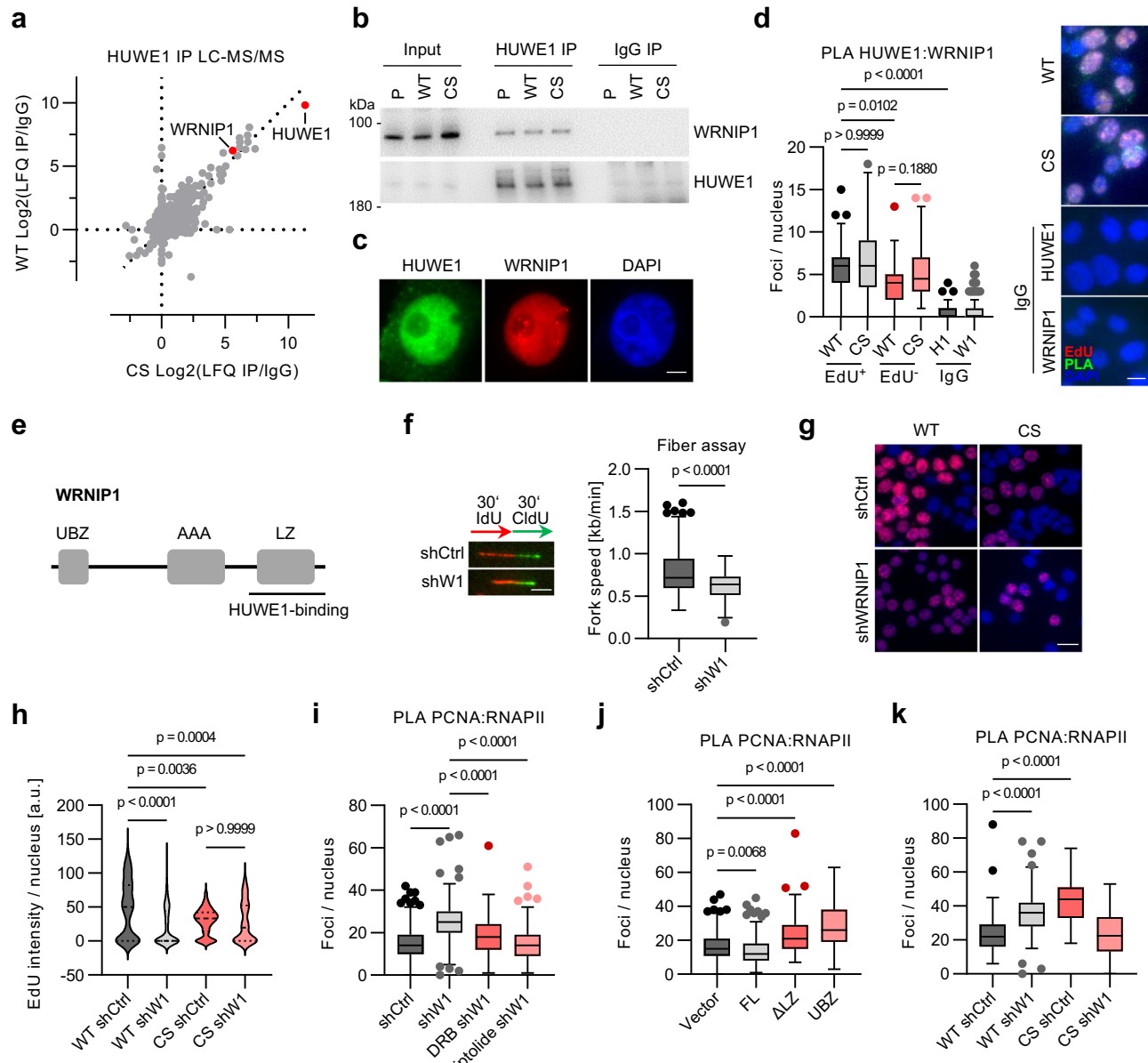

**Fig. 2 | HUWE1 interacts with WRNIP1. a** LC-MS/MS analysis of HUWE1 immuno-precipitates in HUWE1-WT and HUWE1-CS cells. **b** Validation of WRNIP1 as an interactor f HUWE1 by immunoprecipitation (IP) and immunoblotting (*n* = 3). **c** Immunofluorescence staining of HUWE1 and WRNIP1 in HCT116 cells (*n* = 3). Scale bar: 5 μm. **d** PLA assay with HUWE1 and WRNIP1 antibodies in WT and CS cells after pulse-labeling with 25 μM EdU for 30 min. PLA foci were quantified and are displayed for EdU-positive (EdU+) and EdU-negative (EdU−) cell populations and for cells stained solely with either the HUWE1-IgG (H1) or the WRNIP1-IgG (W1). From left, *n* = 128, 161, 107, 92,199, 214 cells. Scale bar: 10 μm. **e** Mapping of the WRNIP1 domain required for HUWE1 binding. UBZ: Ubiquitin binding zinc finger; LZ: Leucine zipper; AAA: AAA+ ATPase domain. See also Supplementary Fig. 2e, f. **f** DNA fiber assay in HCT116 cell expressing either shCtrl or shWRNIP1. Example images (left) and quantification (right) are shown (*n* = 125 fibers per group). *P*-values were determined using the non-parametric, two-tailed Mann–Whitney test. Scale bar:

4 μm. **g** EdU incorporation assay in WT or CS cells expressing shCtrl or shWRNIP1 (shW1). Scale bar: 20 μm, *n* = 2. **h** Quantification of the EdU incorporation assay shown in (**g**). From left, *n* = 192,165,184,115 cells. **i** Quantification of RNAPII-PCNA PLA foci in control and WRNIP1-depleted cells treated with 100 μM DRB or 100 nM triptolide for 6 h (≥100 cells per group). **j** RNAPII-PCNA PLA foci quantification of cells expressing full-length (FL) HA-WRNIP1, WRNIP1 lacking the leucine zipper domain (ΔLZ) or only the WRNIP1 UBZ domain (UBZ). See also Supplementary Fig. 2f. From left, *n* = 190,185,176,179 cells. **k** Quantification of RNAPII-PCNA PLA in HUWE1-WT and HUWE1-CS cells, expressing shCtrl or shWRNIP1. From left, *n* = 177,140,137,132 cells. **d**, **h**–**k** Boxplots show median±quartiles with whiskers ranging up to 1.5-fold of the inter-quartile range. *P*-values were determined using Kruskal–Wallis test followed by Dunn's multiple comparison. Source data are provided as a Source Data file.

histone H2AX, which marks the sites of DNA damage, but is also induced in the vicinity of stalled replication forks[38–40] (Fig. 4a; Supplementary Fig. 4a). Depletion of WRNIP1 in HUWE1-WT cells also induced phosphorylation of ATM targets, whereas overexpression of WRNIP1 in HUWE1-WT and HUWE1-CS cells had the opposite effect (Fig. 4a; Supplementary Fig. 4b). Depletion of WRNIP1 did not increase pATR levels, and neither depletion of WRNIP1 or HUWE1 mutation

alone induced pS4/8, a marker of collapsed replication forks, consistent with the idea that HUWE1 and WRNIP1 can independently bind to and stabilize replication forks (Fig. 4a; Supplementary Fig. 2h–j).

Neutral comet assay showed no increase in levels of DNA breakage in shWRNIP1 cells and a small increase (1.5-fold) in HUWE1-CS cells, compared with HUWE1-WT cells (Fig. 4b). In contrast, depletion of WRNIP1 in HUWE1-CS cells, which reduces TRCs

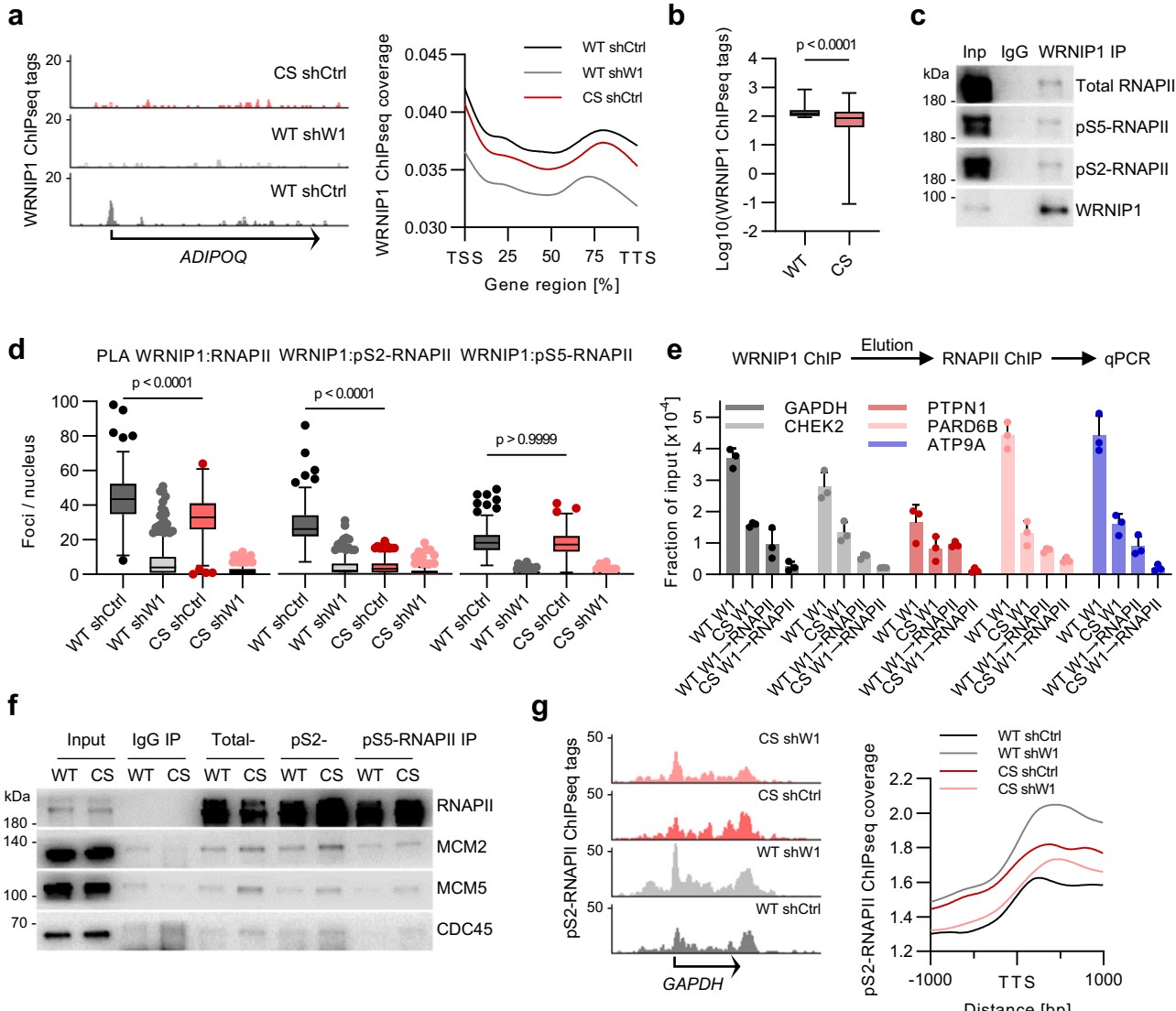

**Fig. 3 | HUWE1 controls association of WRNIP1 with elongating RNAPII.**
**a** Representative genome browser tracks (left) and quantification (right) of WRNIP1 ChIPseq experiments. WRNIP1 ChIPseq tags were quantified at genic regions of all genes and displayed as scaled metagenes. **b** WRNIP1 ChIPseq normalized tag counts for the top 1000 WRNIP1-bound genes in HUWE1-WT cells compared with HUWE1-CS cells ($n = 1000$ genes). Boxplots represent median±quartiles with whiskers ranging from minimum to maximum values. Significance was determined using a paired, two-tailed $t$ test. **c** Immunoprecipitation (IP) analysis with WRNIP1 in benzonase-treated HCT116 lysates ($n = 3$). **d** PLA with antibodies to WRNIP1 and RNAPII ($n = 202, 221, 230, 182$ cells), pS2-RNAPII ($n = 179, 216, 232, 207$ cells) and pS5-RNAPII ($n = 181, 202, 252, 182$ cells) in HUWE1-WT and HUWE1-CS cells

expressing shWRNIP1 (shW1) or a shCtrl. Boxplots show median±quartiles with whiskers ranging up to 1.5-fold of the inter-quartile range. *P*-values were determined using Kruskal-Wallis test followed by Dunn's multiple comparison. **e** Re-ChIP experiment consisting of a first ChIP with WRNIP1 antibodies followed by elution of protein-DNA complexes and a second ChIP with RNAPII antibodies. Purified DNA was analyzed by qPCR with the indicated primer pairs ($n = 3$, mean ± SD). **f** Immunoprecipitation of total, pS2-, or pS5-RNAPII in formaldehyde-crosslinked WT and CS cells ($n = 2$). **g** ChIPseq analysis of pS2-RNAPII in the indicated cell lines. Representative genome browser tracks (left) and quantification of pS2-RNAPII ChIPseq tag coverage at all transcription termination sites (right). Source data are provided as a Source Data file.

(Fig. 2k), strongly elevated (3.6-fold) DNA DSBs (Fig. 4b). Analysis of genome-wide distribution of DNA DSBs using DSB-capture and sequencing[41] showed a strong increase in DNA breakage in WRNIP1-deficient HUWE1-CS cells compared to the other three cell lines (Fig. 4c). Around 25 percent of DSB peaks localized to gene bodies and intragenic sites, however most DSBs were promoter-proximal (Fig. 4c; Supplementary Fig. 4c, d), distinct from regions, at which pS2-RNAPII accumulated in HUWE1-CS or in shWRNIP1 cells (Fig. 3g). On average, DSB capture signal correlated well with WRNIP1 and pS2-RNAPII binding (Fig. 4d, Supplementary Fig. 4e). Consistently, genes harboring breaks showed a slightly higher expression level (normalized RNA-seq tags) compared to all genes (Supplementary Fig. 4f). However, there was no systematic difference in expression

of these genes upon mutation of HUWE1 or depletion of WRNIP1 (Supplementary Fig. 4g).

ATM inhibition strongly reduced pH2AX and pKAP1 levels in HUWE1-CS and shWRNIP1 cells, but had a marginal effect in the double mutant cells (Fig. 4e). This result suggested that phosphorylation events in the double mutant cells are primarily dependent on a different kinase, most likely DNA-PK, which can target H2AX and KAP1 at DSB sites[42–46]. The induction of DNA damage in WRNIP1-depleted HUWE1-CS cells was accompanied by a strong decrease in cell viability (Fig. 4f).

**HUWE1 and WRNIP1 limit ATM activation at RNAPII**
Like TRCs, phosphorylation of ATM and ATM targets in HUWE1-CS cells was diminished by treatment with CDK9 inhibitor AZD4573 or by

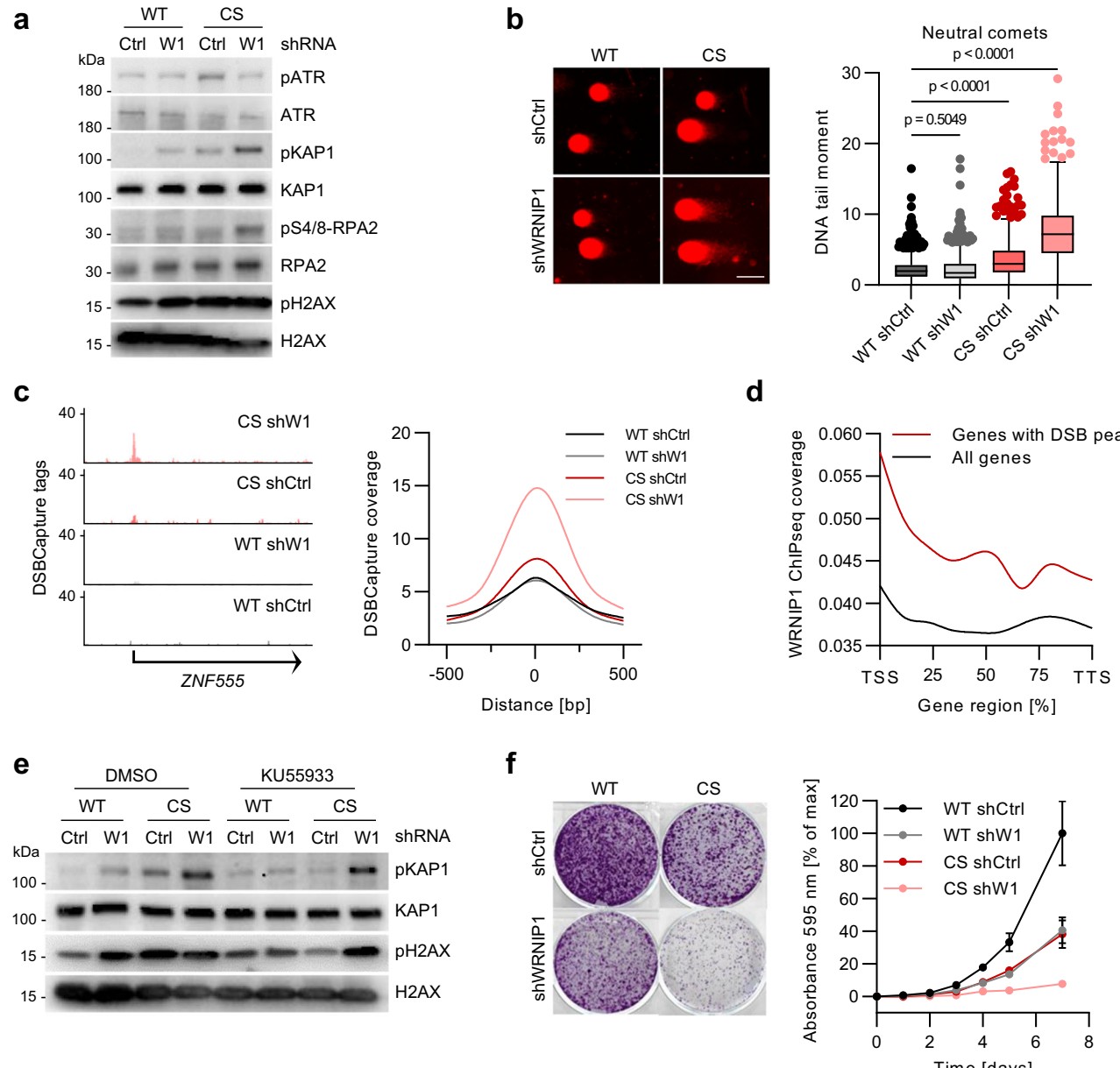

**Fig. 4 | HUWE1 and WRNIP1 suppress ATM signaling. a** Immunoblotting analysis of HUWE1-WT and HUWE1-CS cells expressing shWRNIP1 or shCtrl (*n* = 3). **b** Neutral comet assay of HUWE1-WT and HUWE1-CS cells with or without WRNIP1 depletion (*n* = 1395, 1045, 779, 660 comets). Boxplots (right) show median±quartiles with whiskers ranging up to 1.5-fold of the inter-quartile range. P-values were determined using Kruskal–Wallis test followed by Dunn's multiple comparison. Scale bar: 50 μm. **c** Representative tracks (left) and quantification of DSBCapture tag density (right) in HUWE1-WT and HUWE1-CS cells expressing shWRNIP1 or shCtrl. **d** Metagene analysis of WRNIP1 ChIPseq coverage for all genes and genes with DSBCapture peaks in HUWE1-WT cells. **e** Immunoblotting analysis of WT and CS cells expressing shCtrl or shWRNIP1, treated with 2.5 μM ATM inhibitor KU55933 or DMSO for 2 h (*n* = 3). **f** Analysis of proliferation of HUWE1-WT or HUWE1-CS cells expressing shCtrl or shWRNIP1 by crystal violet staining (*n* = 3, mean ± SD). Source data are provided as a Source Data file.

triptolide (Fig. 5a; Supplementary Fig. 5a, b), suggesting that elongating RNAPII promotes ATM activation and leading us to analyze the underlying mechanisms. Previous studies showed that RNAPII readily interacts with and colocalizes on chromatin with the MRN complex, which recruits ATM to the sites of DNA DSBs (Fig. 5b) (Salifou et al, 2021; Sharma et al, 2021). Our mass spectrometry analysis of pS2-RNAPII interactome in mouse embryonic fibroblasts identified RAD50 and MRE11 (Supplementary Fig. 5c), suggesting that recruitment of MRN is general feature of elongation-competent RNAPII. RAD50 knockdown suppressed phosphorylation of ATM targets in HUWE1-CS cells (Fig. 5c), showing that the MRN complex mediates ATM activation upon mutation of HUWE1.

Immunoprecipitation experiments revealed a robust interaction of pS2-RNAPII with RAD50 and with MRE11 in HCT116 cells (Fig. 5d). Mutation of HUWE1 and depletion of WRNIP1 did not strongly affect the MRN-RNAPII interaction (Supplementary Fig. 5d), but stimulated ATM recruitment to RNAPII (Fig. 5e, f). RNAPII was also readily co-precipitated with antibodies to active, S1981-phosphorylated ATM in HUWE1-CS cells (Supplementary Fig. 5e). Treatment with triptolide reverted the increase in ATM-RNAPII interaction in HUWE1-CS cells (Fig. 5f), in line with the effect on TRCs and ATM signaling (Fig. 1e; Fig. 5a). Knockdown of RAD50 also diminished ATM recruitment by RNAPII in HUWE1-CS cells, suggesting that mutation of HUWE1 promotes ATM association with RNAPII via the MRN complex (Supplementary Fig. 5f).

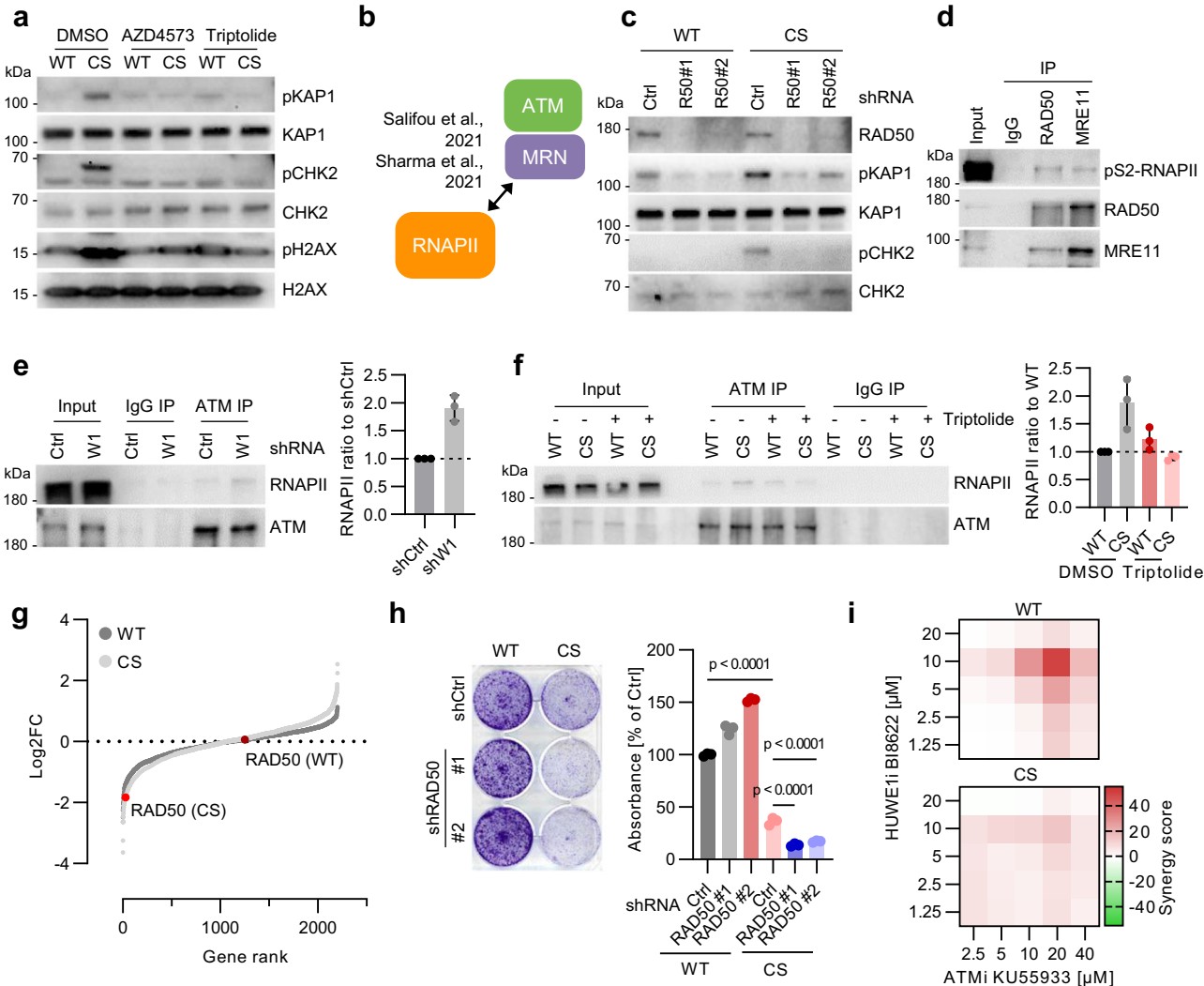

**Fig. 5 | HUWE1 and WRNIP1 limit ATM activation at RNAPII. a** Immunoblotting analysis of HUWE1-WT and HUWE1-CS cells treated with 20 nM AZD4573 or 20 nM triptolide for 6 h ($n = 4$). **b** Scheme of RNAPII-MRN-ATM interaction. See also Supplementary Fig. 5c. **c** Immunoblot analysis of HUWE1-WT and HUWE1-CS cells expressing shCtrl or shRAD50 ($n = 3$). **d** Immunoprecipitation (IP) of RAD50 and MRE11 in benzonase-treated lysates of HCT116 cells ($n = 2$). **e** Immunoprecipitation of ATM in benzonase-treated lysates of shWRNIP1 or shCtrl cells. A representative image (left) and densitometry-based quantification (right, $n = 3$, mean ± SD) are shown. **f** Immunoprecipitation of ATM in benzonase-treated lysates using HUWE1-WT and HUWE1-CS cells treated with DMSO (−) or 20 nM triptolide (+) for 4 h.

A representative image (left) and densitometry-based quantification (right, $n = 3$, mean ± SD) are shown. **g** CRISPR-KO screen using a focused sgRNA library targeting genes within GO terms "Cell cycle", "DNA replication" and "DNA repair" in HUWE1-WT and HUWE1-CS cells. **h** Crystal violet staining (left) and quantification (right, $n = 3$, mean ± SD) of HUWE1-WT or HUWE1-CS cells expressing shRAD50 or shCtrl after 7 days of unperturbed growth. *P*-values were determined with one-way ANOVA followed by Tukey's multiple comparison. **i** Synergy of ATM and HUWE1 inhibitors in killing of HUWE1-WT and HUWE1-CS cells determined by WST-8 assay after 4 d of treatment. Source data are provided as a Source Data file.

Notably, we identified RAD50 as a gene selectively required for the viability of HUWE1-CS cells in a functional sgRNA library screen (Fig. 5g). To validate this result, we expressed two independent shRNAs targeting RAD50 in HUWE1-WT and HUWE1-CS cells using lentiviral transduction. Depletion of RAD50 significantly inhibited proliferation of HUWE1-CS cells but not of HUWE1-WT cells (Fig. 5h). Consistently, mutation of HUWE1-sensitized cells to ATM inhibitor KU-55933 (Supplementary Fig. 5g) and acute inactivation of HUWE1 with BI-8622 synergized with ATM inhibition in HUWE1-WT but not in HUWE1-CS cells (Fig. 5i). Together, these results indicate that activation of ATM at TRCs upon inactivation of HUWE1 limits DNA damage and supports cell viability.

## HUWE1 and WRNIP1 control RNAPII-dependent ATM activation upon replicative stress

We then studied the requirement for WRNIP1 and HUWE1 in response to hydroxyurea (HU) treatment, which depletes the cellular

deoxyribonucleotide pool and induces robust replicative stress (Petermann et al, 2010). Consistent with published studies[36,47], PLA and immunoprecipitation experiments showed a rapid WRNIP1 recruitment to the replisome in response to HU (Fig. 6a, Supplementary Fig. 6a). As reported previously[37], this interaction required the UBZ domain of WRNIP1 (Supplementary Fig. 6b). Strikingly, WRNIP1 recruitment to forks was paralleled by a rapid drop in WRNIP1-RNAPII binding (Fig. 6a, Supplementary Fig. 6a), indicating that the replisome and RNAPII provide alternate binding sites for WRNIP1.

HU triggered rapid induction of TRCs for both pS2- and total RNAPII, which was abolished by transcription inhibitors AZD4573 and triptolide (Fig. 6b, Supplementary Fig. 6c, d). The incidence of TRCs did not further increase in HUWE1-CS or shWRNIP1 cells upon exposure to HU (Supplementary Fig. 6e), suggesting that WRNIP1 dissociation from pS2-RNAPII underlies induction of TRCs upon replicative stress. In line with previous studies (Kanu et al, 2016), HU

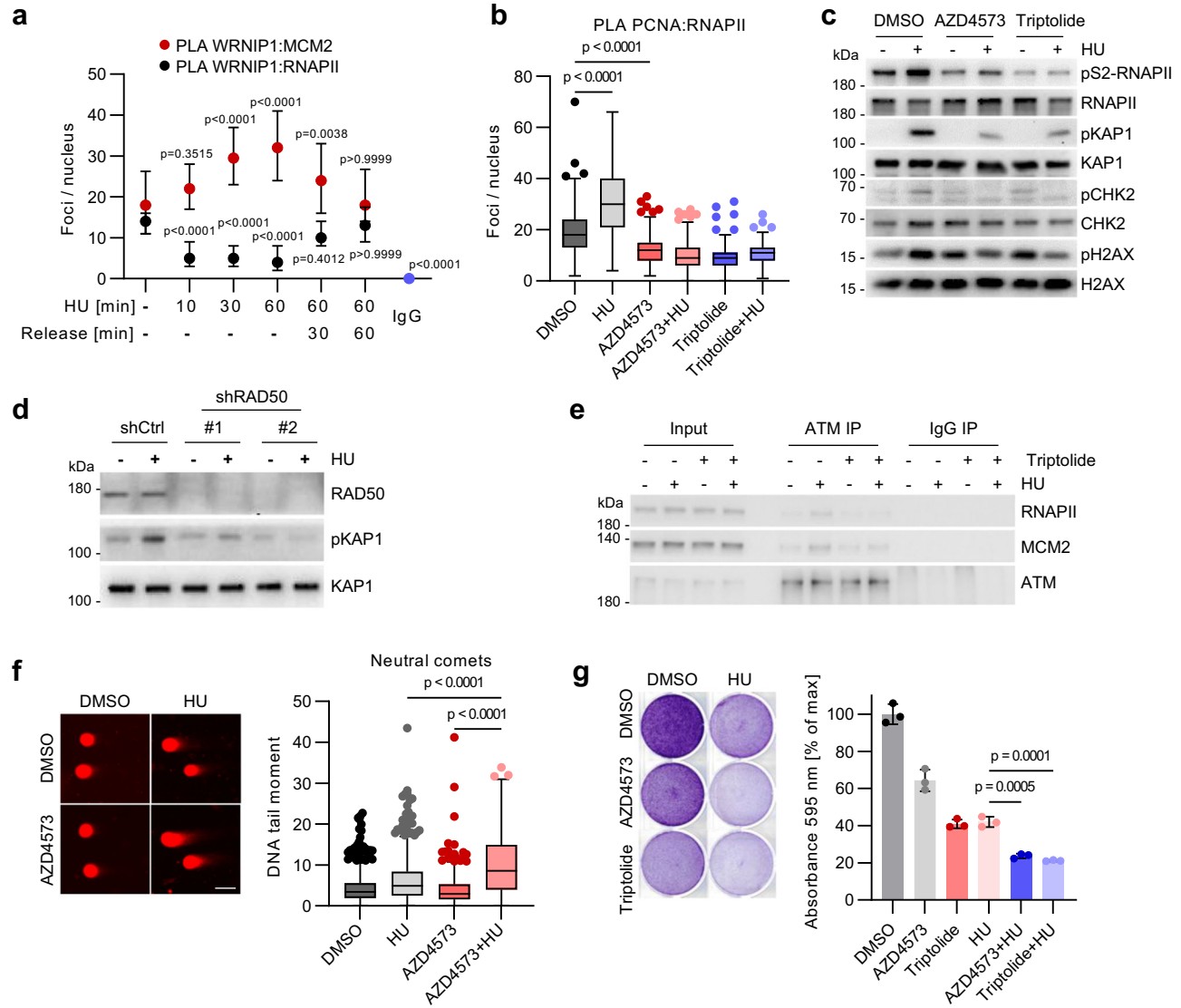

**Fig. 6 | HUWE1 and WRNIP1 control RNAPII-dependent ATM activation upon replicative stress. a** PLA with antibodies against WRNIP1 and MCM2 ($n$ = 190, 167, 132, 140, 175, 172, 207 cells) or WRNIP1 and RNAPII ($n$ = 171, 145, 130, 140, 137, 161, 207 cells) in HCT116 cells treated with 1 mM hydroxyurea (HU) or released from HU for the indicated time. The graph shows median±quartiles for counts of proximity pairs in nuclei. P-values were determined using Kruskal–Wallis test followed by Dunn's multiple comparison. Displayed $p$-values correspond to the comparison of the respective sample to the non-treatment control. **b** PLA with antibodies to PCNA and RNAPII in cells with 2 h pre-treatment using 20 nM AZD4573 or 100 nM triptolide followed by the addition of 1 mM HU for 4 h ($n$ = 187, 160, 195, 223, 198, 171 cells). **c** Immunoblot analysis of HCT116 cells treated with 1 mM HU for 4 h or in combination with 2 h pre-treatment using 20 nM AZD4573 or 100 nM triptolide

($n$ = 3). **d** Immunoblot analysis of HCT cells expressing shCtrl or shRAD50 after treatment with 1 mM HU for 2 h ($n$ = 2). **e** Immunoprecipitation (IP) of ATM in cells treated with 1 mM hydroxyurea or in combination with 20 nM triptolide for 4 h ($n$ = 2). **f** Neutral comet assay in HCT116 cells treated with DMSO, 1 mM HU alone or in combination with 20 nM AZD4573 for 6 h, followed by a 24 h release phase. Scale bar: 50 μm. From left, $n$ = 585, 517, 376, 320 cells. **g** Crystal violet staining of HCT116 cells treated with 1 nM AZD4573 or 2.5 nM triptolide for 3 h followed by addition of 250 μM hydroxyurea (HU) for 4 days. P-values were determined using one-way ANOVA followed by Tukey's multiple comparison. **b**, **f** Boxplots show median ±quartiles with whiskers ranging up to 1.5-fold of the inter-quartile range. *P*-values were determined using Kruskal–Wallis test followed by Dunn's multiple comparison. Source data are provided as a Source Data file.

stimulated phosphorylation of ATM targets KAP1 and CHK2 in HUWE1-WT cells, which was diminished by AZD4573 and triptolide (Fig. 6c), suggesting that RNAPII elongation is required for ATM activation upon HU. Depletion of RAD50, which prevents RNAPII-ATM interaction, also diminished phosphorylation of KAP1 and of 53BP1 (Fig. 6d; Supplementary Fig. 6f), indicative of impaired ATM activation. In contrast, pH2AX which also marks sites of DNA DSBs, was increased in HU-treated shRAD50 cells (Supplementary Fig. 6f). Transcription inhibitors diminished HU-induced recruitment of ATM to RNAPII (Fig. 6e, Supplementary Fig. 6g), mimicking the results obtained in HUWE1-CS cells (Fig. 5f). Importantly, HU-induced association of ATM to MCM2 was also impaired by transcription inhibitors, indicating that RNAPII

promotes activation of ATM at replication forks stalled at TRCs. Interestingly, MRE11 was readily detectable in WRNIP1 precipitates in unstressed cells but not in HU-treated cells (Supplementary Fig. 6h), raising the possibility that WRNIP1 antagonizes ATM recruitment by RNAPII-bound MRN complex. We then treated cells for 24 h with HU alone or in combination with AZD4573 or triptolide and analyzed the effects on DNA damage. Immunoblotting analysis showed that co-treatment with transcription inhibitors impaired the recovery of pH2AX following release from the treatment (Supplementary Fig. 6i), indicative of accumulation of DNA damage in the double treated cells. Indeed, neutral comet assays revealed a strong increase in DNA breakage in cells co-treated with HU and AZD4573 compared with

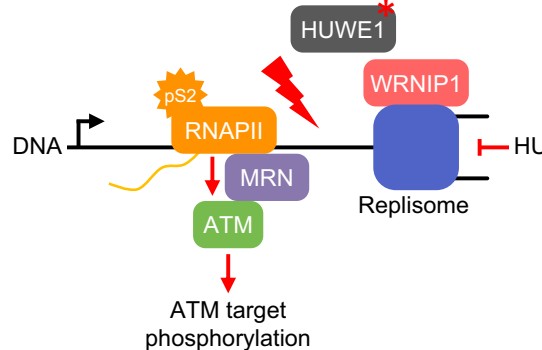

**Fig. 7 | Proposed model of HUWE1 and WRNIP1 dependent coordination of transcription and replication.** HUWE1 binds to WRNIP1 and both proteins prevent transcription-dependent TRCs. HUWE1 promotes the association of WRNIP1 with elongation-competent pS2-RNAPII. Upon induction of replication stress or in HUWE1-CS cells, WRNIP1 translocates from RNAPII to stalled replication forks, leading to stabilization of TRCs and recruitment of ATM to RNAPII via the MRN complex. Stabilization of the collision state induces ATM signaling and limits DNA breakage.

either single treatment (Fig. 6f). Pretreatment with AZD4573 and triptolide impaired cell survival after exposure to HU (Fig. 6g, Supplementary Fig. 6j), in agreement with earlier observations[48]. We concluded that RNAPII-localized activation of ATM upon replicative stress promotes DNA repair and cell survival.

## Discussion

In this study, we show that ubiquitin ligase HUWE1 promotes association of the ATPase WRNIP1 with elongating RNAPII, which diminishes transcription-replication conflicts (TRCs) and limits ATM activation during unperturbed cell cycle (Fig. 7). Inhibition of HUWE1 induces dissociation of WRNIP1 from RNAPII, induces TRCs and localized ATM activation on RNAPII. TRCs also accumulate upon hydroxyurea-induced replicative stress to promote WRNIP1 dissociation from RNAPII, activate ATM and promote DNA repair.

DNA damage response signaling, characterized by the activation of ATM and ATR kinases, is triggered by DNA lesions to promote DNA repair[49–51]. The best-understood signal for ATM activation is DNA DSBs, which are recognized by the MRN complex that includes the exonuclease MRE11 and an ABC ATPase RAD50[52]. MRN recruits ATM, which phosphorylates a range of proteins at the sites of damage, including histone H2AX, transcriptional regulator KAP1 and the effector kinase CHK2[53]. The ATM and ATR kinases are also activated upon stalling of DNA replication forks, facilitating fork stabilization and the restart of DNA synthesis during recovery from stress[18,51]. One proposed mechanism for ATM recruitment at the replisome involves the ATPase WRNIP1[35], which stabilizes stalled replication forks and facilitates the restart of DNA synthesis[36,37,54]. In line with these studies, we find that WRNIP1 recruitment to the replisome is stimulated by replicative stress. However, we show that WRNIP1 is not essential for ATM activation. Moreover, the depletion of WRNIP1 alone induces ATM signaling, suggesting that WRNIP1 limits ATM activation during unperturbed cell cycle.

We show that in the absence of stress, WRNIP1 binds RNAPII and broadly associates with transcriptionally active chromatin. As for the replisome[37], association with RNAPII requires WRNIP1 Ub-binding domain, suggesting a simple mechanism for alternate WRNIP1 recruitment. The dissociation of WRNIP1 from RNAPII and recruitment to replication forks is evoked upon global fork stalling (e.g., under hydroxyurea treatment) but may also occur locally and transiently during normal cell cycle to coordinate DNA replication with RNAPII transcription[55–57].

WRNIP1 association with elongating RNAPII requires ubiquitin ligase HUWE1, which we identify as a major binding partner of WRNIP1.

Mutation of HUWE1 inhibits WRNIP1 association with RNAPII but stimulates its interaction with replication forks, mimicking HU-induced phenotype. HUWE1 may promote WRNIP1 RNAPII binding via ubiquitination of an RNAPII-associated protein, so that catalytic inactivation of HUWE1 function would cause WRNIP1 dissociation. Alternatively, stalled forks in HUWE1-CS cells present a high-affinity binding site for WRNIP1, such as ub-PCNA[35,58,59]. WRNIP1 likely interacts with stalled forks across the genome but may preferentially bind forks stalled at TRCs (e.g., induced by HUWE1 mutation), since they juxtapose WRNIP1 binding sites on RNAPII and the replisome. Based on previous studies that show a role of WRNIP1 in fork stabilization (Leuzzi et al, 2016), we propose that WRNIP1 can stabilize replication forks that collide with RNAPII in HUWE1-CS cells. Supporting this idea, the depletion of WRNIP1 in HUWE1-CS cells reverts the increase in TRC levels.

Curiously, depletion of WRNIP1 in HUWE1-WT cells also induces TRCs, which are dependent on HUWE1 catalytic function. Since either mutation of HUWE1 or loss of WRNIP1 induce conflicts, we propose that HUWE1 activity and WRNIP1 jointly promote TRC resolution but redundantly stabilize forks during collisions. HUWE1 and WRNIP1 may regulate a common interaction partner such as PCNA or histone H1, which are both implicated in the resolution of TRCs[28,29,58,60,61]. For example, HUWE1 may promote ubiquitination of such a protein in the absence of WRNIP1, whereas WRNIP1 binding can functionally compensate for this modification in HUWE1-CS cells. Inactivation of both proteins rescues the increase in TRCs, suggesting that in the double mutant cells replication forks cannot arrest in the vicinity of RNAPII complexes.

Our ChIP-seq experiments show an accumulation of pS2-RNAPII in the 3' regions of a large group of genes upon mutation of HUWE1 or depletion of WRNIP1. As this accumulation correlates with the incidence of TRCs, as determined by the PLA assays, we hypothesize that these genomic regions may represent putative TRC sites. On the other hand, DSB capture experiments show that DSBs in WRNIP1-depleted HUWE1-CS cells predominantly occur at promoters, distant from the sites of pS2-RNAPII accumulation in single mutant cells. One possible interpretation of these data is that in cells lacking both functional HUWE1 and WRNIP1, replication forks do not stall during collisions with RNAPII but progress further and collapse at different sites. Although the mechanism of DSB formation in WRNIP1-depleted HUWE1-CS cells remains to be investigated, our data clearly demonstrate that TRCs are not necessarily accompanied by physical DNA damage.

We hypothesize that TRCs can provide a controlled mechanism for stalling of replication forks that facilitates fork stabilization and

restart during recovery from stress. Our data suggest that a key element of this mechanism is the induction of ATM signaling, which was shown to stabilize forks and promote their restart[18,51]. Both TRCs and ATM activation are dependent on elongating RNAPII and we propose that RNAPII association with the MRN complex underlies this requirement. Previous work has shown that RNAPII can associate with the MRN complex at DNA DSBs[62,63] to promote the synthesis of long non-coding RNAs that facilitate DNA repair. However, MRN also globally colocalizes with RNAPII on chromatin during unperturbed cell cycle[64]. In accord, we find that RNAPII readily associates with MRN in unstressed cells, poised for ATM recruitment and activation. WRNIP1 also associates with RNAPII in unstressed cells but preferentially binds to replication forks upon mutation of HUWE1, when the incidence of TRCs increases. Therefore, we speculate that recruitment of WRNIP1 to the replisome can provide a simple mechanism to trigger ATM recruitment to RNAPII, if for example, WRNIP1 sterically interferes with ATM binding to the MRN complex.

We show that TRCs are rapidly induced under hydroxyurea-induced replicative stress and thus may be instrumental for the induction of the ATM signaling at stalled forks. RNAPII inhibitors diminish hydroxyurea-induced TRCs and ATM signaling and also compromise DNA repair and cell survival following prolonged replicative stress, in line with the previous observations[48,53]. Why is RNAPII important for the repair of replication-associated DNA damage? First, RNAPII may "simply" sustain ATM signaling at levels required for efficient DNA repair. Second, RNAPII may play a more specific role in the stabilization of proximally stalled forks. This could involve spatial coupling of active ATM and RNAPII-associated factors important for fork stabilization, remodeling and DNA repair. For example, BRCA2—the key mediator protein in HR-mediated DNA repair and an ATM substrate—broadly associates with RNAPII and is critical for fork stabilization[65,66]. Analogously, other RNAPII-associated factors, including DNA helicases (e.g., RECQ5 and DHX9) or spliceosomal RNA helicases may be activated by ATM to promote resolution of R loops, stabilize and remodel replication forks for efficient restart of DNA synthesis[19,67–69].

Our data thus suggest that contrary to their common perception as genotoxic events, TRCs may be important for the maintenance of genome stability. In particular, TRCs that engage elongating RNAPII appear to be an integral element of cellular response to drug-induced replicative stress. Blockade of RNAPII elongation dampens replicative stress-induced ATM activation, leads to the accumulation of DNA DSBs and impairs cell survival. This mechanism may also be instrumental under oncogene-induced replicative stress, as indicated by sensitivity of MYC-overexpressing cells to transcription inhibitors[70,71]. These findings lead to a simple model of coordinated transcription, DNA replication and ATM signaling (Fig. 7) and provide a rationale for the development of combinatorial antineoplastic therapies.

## Methods

### Cell culture
Cells (HCT116; a kind gift of Martin Eilers) were cultured in Dulbecco's Modified Eagle's Medium with high glucose (DMEM; Gibco) supplemented with 10% (v/v) fetal bovine serum (FBS; PAN-Biotech), 1% (v/v) penicillin-streptomycin, 1% (v/v) non-essential amino acids and 50 μM 2-mercaptoethanol. Cells were maintained in a humidified atmosphere at 37 °C in 7.5% $CO_2$. Cells were routinely tested for mycoplasma contamination by PCR and the identity of HCT116 cells confirmed via STR typing.

### Cell viability assays
Cell growth was assessed by crystal violet staining of culture dishes after indicated durations followed by absorbance measurements at 595 nm using an Infinite M plex plate reader (Tecan).

The WST-8 assay (PromoKine) was performed according to the manufacturer's instructions. The synergism of combinatorial treatments was assessed with Synergy Finder 2.0[72].

### Immunoblotting
Cells were lysed with 4x Sample Buffer (250 mM Tris-HCl pH 6.8, 4% (v/v) SDS, 25% (v/v) glycerol, 100 mM dithiothreitol, 0.02% (w/v) bromophenol blue) supplemented with phosphatase and protease inhibitor cocktails (Sigma) and boiled for 10 min at 95 °C. SDS-PAGE was performed using 9% Bis-Tris acrylamide gels with MOPS running buffer for 1 h at 125 V followed by wet transfer to PVDF membranes in transfer buffer (25 mM Tris, 192 mM glycine, 10% (v/v) methanol) for 2 h with 125 V at 4 °C. Membranes were blocked using 5% BSA (w/v) in TBST (50 mM Tris-HCl pH 7.6, 150 mM NaCl, 0.1% Tween20) for at least 1 h followed by primary antibody incubation (1:1000 in TBST) overnight at 4 °C. Membranes were then incubated with HRP-linked secondary antibodies (Cell signaling; 1:7500 in TBST) for at least 2 h prior to development using Immobilon Western HRP Substrate (Merck) with a ChemiDoc MP imaging system (Biorad) and the ImageLab (version 5.2.1) software. Details of antibodies are provided in Table 1. Scans are provided in the source data file.

### Immunofluorescence and proximity ligation assay
HCT116 cells were seeded on sterilized coverslips at least 24 h prior to fixation. Reaching about 70% confluence, coverslips were washed with PBS and fixed using 1% paraformaldehyde (PFA) in PBS for 10 min. Cell membranes were permeabilized using 0.3% (v/v) TritonX-100 in TBS for 5 min followed by blocking with 5% BSA (w/v) in TBST for 30 min. Primary and respective fluorophore-linked secondary antibodies were subsequently added to the slides with a 1:200 dilution in blocking buffer for 2 h each. Coverslips were mounted on slides using VECTA-SHIELD HardSet Antifade Mounting Medium with DAPI (Biozol) and stored at 4 °C until picture acquisition with an Olympus DP80 mounted to an Olympus BX63 using the cellSens Dimension (version 1.17) software.

For proximity ligation assays (PLA)[73], PFA fixed slides were additionally fixed with 90% (v/v) methanol at −20 °C overnight and rehydrated in PBS before blocking, permeabilization and primary antibody incubation as above. The two PLA probes (mouse plus and rabbit minus; Sigma) were diluted 1:20 in blocking buffer, added to the slides for 1 h at 37 °C and the development was performed using Duolink® In Situ Detection Reagents Red (Sigma) according to the manufacturer's instructions. For proper quantification of the dot-shaped PLA signals, images were acquired as Z-stacks with at least 5 frames and a Z-increment of 1 μm prior to maximum intensity projection into a single image.

Quantification of nuclear staining intensity and counting of nuclear PLA foci was performed using the free software FIJI/ImageJ (version 1.53 f, https://imagej.net/software/fiji/)[74].

### DNA fiber assay
Desired cells were labeled with 25 μM IdU (5-Iodo-2′-deoxyuridine) and 250 μM CldU (5-Chloro-2′-deoxyuridine) for 30 min each[75]. Approximately 2000 cells were lysed with 200 mM Tris pH 7.5, 50 mM EDTA, 0.5% SDS on a glass slide and tilted at a 15° angle. Spread DNA fibers were fixed with 67% methanol, 33% acetic acid for 10 min and denatured using 2.5 M HCl for 80 min. Slides were blocked with 5% BSA in PBS for 20 min and primary antibodies were added for 2 h. Secondary antibodies (Cell signaling) were diluted in blocking buffer and added for 1 h prior to image acquisition. Fibers were analyzed with FIJI/ImageJ (version 1.53 f).

### EdU-Click reaction
Cultured cells on coverslips were treated with 25 μM 5-Ethynyl-2′-deoxyuridine (EdU) for 30 min, fixed with 4% PFA for 5 min and

## Table 1 | Antibodies used in this study

| Target | Catalog number | Manufacturer | Reactivity | Application |
|---|---|---|---|---|
| 53BP1 | 4937 | Cell Signaling | H Mk | WB, IHC, IF |
| ATM (G-12) | sc-377293 | Santa Cruz | H | WB, IP, IF, IHC(P) and ELISA |
| ATR | 2790 | Cell Signaling | H Mk | WB |
| BrdU (B44) | BD347580 | BD | Species independent | Flow cytometry, Intracellular staining (flow cytometry) |
| BrdU [BU1/75 (ICR1)] | ab6326 | Abcam | Species independent | ICC/IF, Flow Cyt (Intra), IHC-P |
| CDC45 | 11881S | Cell Signaling | H M R Mk | WB, IP, IF |
| CHK2 | 2662T | Cell Signaling | H M R Mk | WB, IP |
| H2AX (938CT5.1.1) | sc-517336 | Santa Cruz | H | WB, IP, IF, and IHC(P) |
| HA-tag (6E2) | 2367 | Cell Signaling | Species independent | WB, IHC, IF, F |
| HA-tag (C29F4) | 3724 | Cell Signaling | Species independent | WB, IP, IHC, IF, F, ChIP |
| Histone H2B (53H3) | 2934 | Cell Signaling | H M R Mk Z | WB |
| Histone H3 (96C10) | 3638 | Cell Signaling | H M R Mk | WB |
| HUWE1 | A300-486A-A | Bethyl | H | WB, IP, IHC |
| KAP1 | 15202-1-AP | Proteintech | H M R | WB, IP, IHC, IF, FC, ChIP, ELISA |
| MCM2 (D7G11) | 3619S | Cell Signaling | H M R Mk | WB, IP, IHC, IF, ChIP |
| MCM5 | ab17967 | Abcam | H | WB, IHC-P |
| MRE11 | NB100-142 | Novus Biologicals | Hu, Mu, Rt, Ch, Ha, Bt, Bv, Ca, Pm, Eq, Fe, Pm | WB, Simple Western, ChIP, ELISA, Flow, IB, ICC/IF, IHC, IHC-Fr, IHC-P, IP, PLA, KD, KO |
| MRE11 (18) | sc-135992 | Santa Cruz | H M R | WB, IP, IF |
| MYC (D3N8F) | 13987 | Cell Signaling | H M R Mk | WB, IF, F, ChIP, C&R |
| PCNA (PC10) | sc-56 | Santa Cruz | H M R Y | WB, IP, IF, IHC(P) and FCM |
| pS139-H2AX (20E3) | 9718 | Cell Signaling | H M R Mk | WB, IHC, IF, F |
| pS139-H2AX | 517348 | Santa Cruz | H M R | WB, IP and IF |
| pS1981-ATM (10H11.E12) | 47739 | Santa Cruz | H M R | WB, IP, IF and IHC(P) |
| pS1981 ATM (EP1890Y) | ab81292 | Abcam | H | Dot blot, Flow Cyt (Intra), WB, IHC-P, IP |
| pS2-RNAPII | 61083 | Active Motif | H M | WB, IP, IF, ChIP, ICC |
| pS2-RNAPII | A300-654A-M | Bethyl | H M | WB, IP, IHC |
| pS25-53BP1 (38.Ser 25) | sc-135748 | Santa Cruz | H M | WB, IP and ELISA |
| pS4/8-RPA2 (E5A2F) | 54762 S | Cell Signaling | H | WB, IF, F |
| pS428-ATR | 2853 T | Cell Signaling | H M R Mk | WB |
| pS5-RNAPII | 39749 | Active Motif | H M | WB, ChIP |
| pS824-KAP1 (BL-246-7B5) | ab243870 | Abcam | H M | IP, IHC-P, WB, ICC/IF |
| pT68-CHK2 | ab3501 | Abcam | H | WB |
| RAD50 (G-2) | sc-74460 | Santa Cruz | H M R | WB, IP, IF and ELISA |
| RAD50 | NB100-154 | Novus Biologicals | Hu, Mu, Ha | WB, Simple Western, IP, ICC/IF |
| RBP1 (RNAPII) (D8L4Y) | 14958 | Cell Signaling | H M R Mk | WB, ChIP |
| RNAPII (4H8) | 101307 | Active Motif | Budding Yeast, C. elegans, Human, Mouse, Rat | WB, IF, ChIP, ICC |
| RPA2 | sc-56770 | Santa Cruz | H M R | WB, IP, IF, IHC(P) and ELISA |
| Vinculin (hVIN-1) | V9131 | Sigma | frog, chicken, mouse, canine, human, bovine, rat, turkey | WB, IF, IHC |
| WRNIP1 (G-2) | 377402 | Santa Cruz | H M R | WB, IP, IF, IHC(P) and ELISA |
| WRNIP1 | A301-389A-T | Bethyl | H | WB, IP, IHC |
| Anti-mouse IgG (Alexa Fluor® 488 Conjugate) | 4408 | Cell Signaling | M | IF |
| Anti-mouse IgG (Alexa Fluor® 555 Conjugate) | 4409 | Cell Signaling | M | IF |
| Anti-rabbit IgG (Alexa Fluor® 488 Conjugate) | 4412 | Cell Signaling | Rabbit | IF |
| Anti-rabbit IgG (Alexa Fluor® 555 Conjugate) | 4413 | Cell Signaling | Rabbit | IF |
| Anti-rat IgG (Alexa Fluor® 488 Conjugate) | 4416 | Cell Signaling | Rat | IF |
| Anti-rabbit IgG, HRP-linked Antibody | 7074 | Cell Signaling | Rabbit | WB |
| Anti-mouse IgG, HRP-linked Antibody | 7076 | Cell Signaling | M | WB |
| Anti-rat IgG, HRP-linked Antibody | 7077 | Cell Signaling | Rat | WB |

For western blot, each primary antibody was diluted 1:1000 and secondary antibodies 1:7500. For immunofluorescence and PLA, both primary and secondary antibodies were diluted 1:200.

permeabilized with 0.3% TritonX-100 in TBS for 5 min. Incorporated EdU was labeled with Cy3 using 2 mM CuSO4, 4 μM Sulfo-Cy3-azide, 100 mM sodium ascorbate for 30 min. Coverslips were then processed and analyzed as other immunofluorescence samples.

## Single-cell gel electrophoresis

Detached cells were embedded in 0.7% low melting point (LMP) agarose and transferred to pre-coated glass slides[76], solidified and coated with a cell free layer of LMP agarose. After lysis with 2.5 M NaCl, 0.1 M EDTA, 10 mM Trizma base (pH 10), 1% N-laurylsarcosine, 0.5% Triton X-100, 10% DMSO overnight, slides were placed in a TAE buffer containing electrophoresis chamber for 1 h at 18 V. DNA tails were fixed with pure ethanol and slides were dried prior to soaking with 2 μg/ml ethidium bromide and image acquisition using a fluorescence microscope. DNA tail properties were analyzed using the OpenComet (v1.3.1) FIJI/ImageJ plugin[77].

## HUWE1 gene replacement

To generate a catalytic mutation of the endogenous HUWE1 gene, HCT116 cells were transfected using Fugene (Promega) with a repair template and two HUWE1 targeting sgRNAs cloned in the PX459 vector as shown previously[30]. The HUWE1 repair template consisted of homology regions resembling genomic positions chrX:53561159-53561889 and chrX:53559367-53560269, the HUWE1 ORF sequence of amino acid residues 4277-4374 (Q7Z6Z7-1; ENST00000342160.7) with either cysteine or serine at position 4341, a P2A site and a blasticidin resistance gene. Single cell clones were propagated after puromycin and blasticidin selection and correct gene replacement was confirmed by PCR and Sanger-sequencing. Oligonucleotide sequences are shown in Table 2.

## WRNIP1 expression and truncation variants

To generate expression vectors of recombinant, HA-tagged WRNIP1, cDNA of HCT116 cells was amplified with primers for either full length (ENST00000380773.9) or truncated variants of WRNIP1. The protein sequence (Q96S55-1) was divided into three parts encompassing the ubiquitin binding domain (UBZ, residues 1–222), the AAA-ATPase domain (AAA, residues 223–444) and the leucine zipper containing domain (LZ, residues 445–665). Amplified cDNA sequences were cloned into pRRL vectors (a gift from Martin Eilers)[78] via NEBuilder® HiFi DNA Assembly Master Mix and transfected together with lentiviral packaging vectors pMD2.G (a gift from Didier Trono, Addgene plasmid #12259) and psPAX2 (a gift from Didier Trono, Addgene plasmid #12260) into LentiX cells using polyethylenimine (PEI). Resulting cell-free, lentivirus containing supernatants were transferred to infect 100,000 HCT116 cells in the presence of 8 μg/ml hexadimethrine bromide for at least 72 h and after hygromycin (200 μg/ml) selection, expression of HA-tagged WRNIP1 and its derivatives was confirmed via western blot.

## RNA interference

Sequences for shRNAs were cloned into constitutively expressing pLKO.1 (a gift from Bob Weinberg, Addgene plasmid # 8453)[79] or doxycycline inducible pLKO tet-on (a gift from Dmitri Wiederschain, Addgene plasmid # 21915)[80] vector backbones using NEBuilder® HiFi DNA Assembly Master Mix or as described[80] and HCT116 cells were infected using a lentivirus as described for protein overexpression. As negative control, we used either empty plasmid backbones or a non-targeting, scrambled shRNA sequence. After puromycin selection with 1 μg/ml for 48–72 h, pLKO tet-on dependent depletion of target proteins was induced with 100 ng/ml doxycycline for at least 4 days prior to any experiment and depletion was confirmed via western blot.

## Immunoprecipitation

HCT116 cells were scraped and lysed in TNT-300 lysis buffer (25 mM Tris pH 7.4, 300 mM NaCl, 1% (v/v) Triton-X100, with protease and phosphatase inhibitor cocktails (Sigma)) for 10 min on ice and diluted 1:1 with TNT-0 to a final NaCl concentration of 150 mM. For Immuno-precipitation of pS2RNAPII for LC-MS/MS, benzonase (Sigma) was added to samples during lysis. Lysates were cleared by centrifugation, adjusted based on protein concentration and transferred to reaction tubes with 30 μl pre-washed protein A or protein G (Thermo) agarose beads. Primary antibodies or respective IgG control antibodies were added and samples were rotated at 4 °C overnight. After washing three times with TNT-150, samples were either analyzed by western blot or by mass spectrometry.

For benzonase treatment, cells were resuspended in benzonase buffer (20 mM Tris, pH7.5, 100 mM NaCl, 2 mM MgCl2, 2 mM CaCl2, 0.2% TX-100) and incubated with 0.2 μl benzonase (Sigma) for 1 h on ice. Nuclei were lysed by the addition of 5 volumes of TNT-150 buffer supplemented with 5 mM EDTA buffer and the IP reactions were performed as described above.

To analyze protein complexes after crosslinking, cells were fixed with 0.2% PFA for 4 min and the reaction was quenched with 200 mM glycine. Scraped cells were sonicated for 3 × 2 min with 30% amplitude, 2 W power and a 45 s ON/ 5 s OFF cycle using the UP200St sonicator (Hielscher). After clearing lysates, immunoprecipitation was performed as described previously and complexes were analyzed via western blot with prolonged boiling to reduce crosslinks between proteins.

## Re-IP and re-ChIP

For each sample, 20 million (re-IP) or 80 million (Re-ChIP) HCT116 cells were crosslinked with 0.2% PFA for 4 min followed by quenching with 200 mM glycine for 4 min. Cells were scraped and lysed in TNT-300 (25 mM Tris, 300 mM NaCl, 1% (v/v) Triton-X100, pH 7.4, with protease/phosphatase inhibitor cocktails) and chromatin was fragmented by sonication for 3 × 2 min with 30% amplitude, 2 W power and a 45 s ON/15 s OFF cycle with an UP200St sonicator (Hielscher) on ice. Lysates were cleared by centrifugation and protein complexes were precipitated with 1–2 μg primary antibodies or respective IgG controls on pre-washed protein A or protein G (Thermo) agarose beads overnight at 4 °C. Captured proteins were eluted from agarose beads in PBS, 2% SDS, 10 mM DTT for 30 min at 30 °C while shaking. Eluates were diluted 1:20 in TNT-300 prior to the second round of immuno-precipitation. Samples were analyzed by immunoblotting for re-IP or DNA was purified as described for ChIPseq. Precipitated DNA was diluted in water and analyzed by qPCR using SYBR Green JumpStart Taq ReadyMix on a BioRad CFX Connect Real-time PCR system following the manufacturer's instructions.

## In-gel digestion

After immunoprecipitation, captured target proteins and interaction partners were eluted with 30 μl 4× sample buffer (250 mM Tris-HCl pH 6.8, 4% (v/v) SDS, 25% (v/v) glycerol, 100 mM dithiothreitol, 0.02% (w/v) bromophenol blue) supplemented with phosphatase and protease inhibitor cocktails (Sigma), boiled for 5 min at 95 °C and loaded to a 9% acrylamide gel for electrophoresis lasting 20 min at 80 V. Sample lanes were cut into 1 mm3 pieces and washed three times for 20 min with 5 mM ammonium bicarbonate (ABC), 50% (v/v) acetonitrile (ACN) and a final wash with pure ACN. Proteins were reduced with 10 mM dithiothreitol, 20 mM ABC for 45 min at 56 °C followed by alkylation with 55 mM iodoacetamide, 20 mM ABC for 45 min. Again, gel pieces were dehydrated as described above and soaked with 12.5 ng/μl trypsin (Promega), 20 mM ABC. Proteins were digested overnight at 37 °C and peptides were extracted by subsequent washes with 50% ACN, 3% trifluoroacetic acid followed by 80% ACN, 0.5% acetic acid and pure ACN for 30 min each. Liquid supernatants were pooled for each sample and organic solvents removed by vacuum centrifugation. Acidified samples (pH 2) were desalted using StageTips[81]. In brief, 2 × 1 mm2 discs of C18 bonded silica (Empore)

**Table 2 | Oligonucleotides used in this study**

| Target | Sequence |
|---|---|
| shWRNIP1 #1 TRCN0000436158 F | CCGGATGATGTGCGAGATGTCATAACTCGAGTTATGACATCTCGCACATCATTTTTTG |
| shWRNIP1 #1 TRCN0000436158 R | AATTCAAAAAATGATGTGCGAGATGTCATAACTCGAGTTATGACATCTCGCACATCAT |
| shWRNIP1 #2 TRCN0000436134 F | CCGGGTGACATTATCTGCAACAAATCTCGAGATTTGTTGCAGATAATGTCACTTTTTG |
| shWRNIP1 #2 TRCN0000436134 R | AATTCAAAAAGTGACATTATCTGCAACAAATCTCGAGATTTGTTGCAGATAATGTCAC |
| shWRNIP1 #3 TRCN0000004527 F | CCGGCGCTGTCGAGTGATTGTTCTTCTCGAGAAGAACAATCACTCGACAGCGTTTTTG |
| shWRNIP1 #3 TRCN0000004527 R | AATTCAAAAACGCTGTCGAGTGATTGTTCTTCTCGAGAAGAACAATCACTCGACAGCG |
| shMYC TRCN0000039642 F | CCGGCCTGAGACAGATCAGCAACAACTCGAGTTGTTGCTGATCTGTCTCAGGTTTTTG |
| shMYC TRCN0000039642 R | AATTCAAAAACCTGAGACAGATCAGCAACAACTCGAGTTGTTGCTGATCTGTCTCAGG |
| shHUWE1 #1F[25] | CCGGCCCGCATGATCTTGAATTTCTCGAGAAATTCAAGATCATGCGGGTTTTTG |
| shHUWE1 #1R[25] | AATTCAAAAACCCGCATGATCTTGAATTTCTCGAGAAATTCAAGATCATGCGGG |
| shRAD50 #1 TRCN0000040104 F | CCGGCGCCTAAAGAACGACATAGAACTCGAGTTCTATGTCGTTCTTTAGGCGTTTTTG |
| shRAD50 #1 TRCN0000040104 R | AATTCAAAAACGCCTAAAGAACGACATAGAACTCGAGTTCTATGTCGTTCTTTAGGCG |
| shRAD50 #2 TRCN0000009838 F | CCGGGAGATTCGTGATCAGATTACACTCGAGTGTAATCTGATCACGAATCTCTTTTTG |
| shRAD50 #2 TRCN0000009838 R | AATTCAAAAAGAGATTCGTGATCAGATTACACTCGAGTGTAATCTGATCACGAATCTC |
| shScrambled F | CCGGAGGCTCGCATGTATACAGACTCGAGTCTGTATACATGCGAGCCTTTTTTG |
| shScrambled R | AATTCAAAAAAGGCTCGCATGTATACAGACTCGAGTCTGTATACATGCGAGCCT |
| sgRNA HUWE1_1[30] | AAGGCCCTGCCCAACTCCGT |
| sgRNA HUWE1_2[30] | CATGCTACTGTTGGCTATCC |
| HA-WRNIP1 cds pRRL F | TGAGTCGGCCGGTGGATCCAATGTACCCTTACGACGTGCCCGACTACGCCGAGGTGAGCGGGCCG |
| HA-WRNIP1 cds pRRL R | GAGGGGCGGATCCGTCGACATCAGCACCTCCTCTGCTTGAAG |
| HA-WRNIP1 trunc1 pRRL F | TGAGTCGGCCGGTGGATCCAATGTACCCTTACGACGTGCCCGACTACGCCCTACAGGGCAAGCCGC |
| HA-WRNIP1 trunc2 pRRL F | TGAGTCGGCCGGTGGATCCAATGTACCCTTACGACGTGCCCGACTACGCTGGGTTGAACGGACTGCAGC |
| HA-WRNIP1 trunc3 pRRL R | GAGGGGCGGATCCGTCGACATCATCGGGCGTCACCGTCACTG |
| HA-WRNIP1 trunc4 pRRL R | GAGGGGCGGATCCGTCGACATCACATCTGTCGGATCTCC |
| HUWE1 clone screen bsr F | GTTGGGATTCGTGAATTGCT |
| HUWE1 clone screen flank R | GGTGTCTTCTTCAGTTTAGTCCTG |
| GAPDH qChIP F | TCCTCTGACTTCAACAGCGAC |
| GAPDH qChIP R | GAGTTGTCAGGGCCCTTTTTC |
| CHEK2 qChIP F | CCACCCTCAGCCAATCAAAAC |
| CHEK2 qChIP R | CCGGGTTCTAAGTTCCGCT |
| PTPN1 qChIP F | GCCCCATGAGCCTTCTGTTA |
| PTPN1 qChIP R | TGCCAAACCGGTCAGAAAGA |
| PARD6B qChIP F | GAGCCCTTCTTCAGGTGCG |
| PARD6B qChIP R | CTCAAAACCCCGCCTACCAC |
| ATP9A qChIP F | CGTGTGAACCACCAGCTATGA |
| ATP9A qChIP R | ATGGCAAGAAAGTTGGCGGG |
| MSL2 qChIP F | CAGGCTTCCCGCATTACACT |
| MSL2 qChIP R | GAGCGGTTCCAGGAGAAAGG |

were placed in a 200 μl pipette tip, activated with methanol and washed with 2% ACN, 1% formic acid (FA) prior to sample loading. Peptides were washed with 0.1% FA and stored at 4 °C.

**TMT-labeling on StageTips**
For labeling of peptides with tandem mass tag (TMT) reagents (Thermo), peptides on StageTips were washed once with HEPES buffer pH 8 and soaked with 4 μl of TMTsixplex™ (Thermo) isobaric label reagent dissolved in ACN for 1 h. Labeled peptides were eluted twice with 20 μl 80% (v/v) ACN, 0.1% FA and the reaction was quenched by adding 0.8 μl of 5% (v/v) Hydroxylamine for 15 min. Organic solvents were evaporated by vacuum centrifugation and peptides were loaded on StageTips as described above under "In-gel digestion".

**NanoLC-MS/MS analysis**
Peptides were separated using a nano-UHPLC (EASY-nLC™ 1200, Thermo) coupled to a Q Exactive™ HFX Hybrid Quadrupole-Orbitrap™ mass spectrometer (Thermo). The mass spectrometer was operated

with the Thermo Xcalibur™ (version 4.3) software. Peptide separation was performed on a column packed with reverse-phase ReproSil-Pur C18-AQ 1.9 μm silica beads (Dr. Maisch GmbH) and a solvent gradient (A: 0.1% FA and B: 80% ACN, 0.1% FA) for 90 min with 200 nl/min. Peptides were ionized via electrospray ionization in positive ion mode and each scan cycle consisted of one full MS scan (60000 resolution, 300–1650 $m/z$ range) followed by selecting the top 12 most abundant precursor ions for further HCD fragmentation with a normalized collision energy of 35%. TMT labeling efficiency and mixing ratio controls were measured using a 36 min gradient and adjusted accordingly.

**Mass spectrometry data analysis**
Data files were processed using the MaxQuant suite (version 1.6.14.0 or version 2.0.3.0)[82] and its integrated peptide search engine Andromeda[83]. Peptide search was performed against the UniProt[84] databases for *Mus musculus* (release_2020-10-07) or *Homo sapiens* (release_2019-12-11) [https://www.uniprot.org/], to which the mutated HUWE1 sequence was added manually. For tryptic cleavage specificity,

2 missed cleavages were allowed and carbamidomethylation (Cys) was set as fixed modification. Further data analysis was performed using Perseus (version 1.6.14)[85].

## RNA sequencing

Total RNA was extracted from HCT116 cells using TRI Reagent® (Sigma) and residual DNA digested using DNAseI (Thermo) for 30 min at 37 °C. RNA was purified and washed via precipitation with 80% ethanol and quantified using a NanoDrop 1000 (Peqlab). The poly(A) mRNA isolation module (NEB) was used to purify mRNA from a total of 0.5 µg RNA. For reverse transcription and NGS library preparation, the NEBNext Ultra II RNA Library Prep Kit for Illumina was used according to the manufacturer's instructions and with a final index PCR with unique pairs of i5 and i7 primers using 12 cycles. DNA fragments and index PCR products were purified with High-Prep™ (MagBio) beads instead of AMPure beads. The DNA content of each library was determined using the Quant-iT™ PicoGreen™ dsDNA Assay-Kit (Thermo) prior to pooling libraries in equimolar ratios for each experiment.

## Chromatin immunoprecipitation (ChIP) and Cut&Run

For each ChIP sample, $4–6 \times 10^7$ HCT116 cells per condition were fixed with 1% PFA for 4 min at room temperature and the reaction was quenched by adding 200 mM glycine for 4 min. Cells were scraped in ice cold PBS and lysed in 200 µl PBS, 0.5% (v/v) TritonX-100, 1 mM dithiothreitol, 5 mM MgCl2, 5 mM CaCl2 and protease/phosphatase inhibitors (Sigma) for 10 min on ice. Chromatin was fragmented with 1 µl micrococcal nuclease (NEB) for 3 min at 37 °C and 20 µl 500 mM EDTA was added to suppress enzyme activity. After diluting samples with 800 µl TNT-300 (25 mM Tris, 300 mM NaCl, 1% (v/v) Triton-X100, pH 7.4, with protease/phosphatase inhibitor cocktails), chromatin was further fragmented by sonication for $3 \times 2$ min with 30% amplitude, 2 W power and a 45 s on/15 s off cycle with an UP200St sonicator (Hielscher) on ice. Lysates were cleared by centrifugation for 10 min at $17,000 \times g$ and 2% of each sample were separated as input. For each immunoprecipitation 1.5–2 µg of target specific antibody was added to the lysate and rotated overnight at 4 °C. To capture proteins of interest, 10 µl magnetic protein A or protein G bead slurry (Thermo) were added per sample and rotated for 2 h. Beads were washed three times with TNT-300, 0.1% SDS and RNA was digested with 1 µl RNAseA (Thermo) in 10 mM Tris pH 8, 1 mM EDTA, 0.5% SDS, 250 mM NaCl for 1 h at 50 °C. Proteins were digested with 2 µl proteinase K (Thermo) at 50 °C for 1 h followed by 65 °C overnight. Fragmented DNA was purified using phenol/chloroform/isoamyl (Roth A156.1), washed and precipitated with 80% ethanol at −20 °C overnight. Input fragment sizes of about 100–800 bp were validated by agarose gel electrophoresis. Indexed libraries were either prepared following the instructions of NEBNext® Ultra™ II DNA Library Prep Kit or according to the "TELP" procedure[86]. In brief, DNA ends were repaired with rSAP (NEB, M0371S) and tailed with dCTP using terminal deoxynucleotidyl transferase (NEB) for 1 h, extended with biotinylated poly-dGTP primers using KAPA2G Robust HotStart ReadyMix (Sigma), captured on streptavidin beads (Thermo) and PCR adapter (NEB) were ligated using Blunt/TA Ligase Master Mix (NEB). Final PCR with indexed, unique i5 and i7 primer pairs was performed using KAPA HiFi HotStart ReadyMix with 22 PCR cycles. All ChIPseq libraries were selected for sizes of 200–1000 bp using agarose gel electrophoresis and subsequent gel extraction (Monarch® DNA Gel Extraction Kit, NEB). The DNA content of each library was determined using the Quant-iT™ PicoGreen™ dsDNA Assay-Kit (Thermo) prior to pooling libraries in equimolar ratios for each experiment.

Cut&Run assays were performed using the CUT&RUN Assay Kit (Cell Signaling)[87] according to the manufacturer's instructions. NGS library preparation was performed using NEBNext® Ultra™ II DNA Library Prep Kit.

## DSB capture and sequencing

Double-strand break sequencing was performed based on the DSBCapture protocol[41]. In brief, 5 million cells were fixed with 0.2% PFA for 4 min and the reaction was quenched by the addition of 200 mM glycine for 4 min. Scraped cells were permeabilized in wash buffer (20 mM HEPES pH 7.5, 150 mM NaCl, 0.5 mM spermidine, 0.1% Triton-X100) for 10 min on ice and bound to Con-A Sepharose (Sigma) in binding buffer (20 mM HEPES, 10 mM KCl, 1 mM CaCl2, 1 mM MnCl2). DNA ends were repaired using Klenow fragment (NEB) in NEB buffer 2 for 30 min at 25 °C followed by A-tailing in dA-tailing buffer (NEB) with Klenow (3′ → 5′ exo-) for 30 min at 37 °C. Biotinylated NEB adaptors were ligated to A-tailed DSBs with T4 DNA ligase (NEB) in the corresponding buffer overnight at 16 °C. Cells were lysed in 10 mM Tris, pH 8.5, 5 mM EDTA, 0.5% SDS, 250 mM NaCl, 1.2 U Proteinase K (Thermo) at 50 °C overnight followed by sonication for 5 min on ice and DNA purification using phenol/chloroform/isoamyl and ethanol. Biotinylated DNA was captured on streptavidin C1 beads (NEB) and washed with 10 mM Tris-HCl pH 8.0, 0.5 mM EDTA, 1 M NaCl, 0.02 % TritonX100. NGS library preparation was performed using NEBNext® Ultra™ II DNA Library Prep Kit.

## Functional sgRNA screen

A pooled sgRNA library targeting about 2300 genes annotated with the gene ontology (GO) terms "DNA replication" (GO:0006260), "DNA repair" (GO:0006281) and "Cell cycle" (GO:0007049) with 4 sgRNA sequences per gene was cloned into the pLentiCRISPR V2 vector backbone following the published strategy[88]. The library was transfected into LentiX cells together with the packaging vectors pMD2.G and psPAX2 as described for protein overexpression. The cell-free, viral supernatants were transferred to 2 replicates of $1.6 \times 10^7$ HCT116 cells each, expressing either the wild-type HUWE1 sequence or HUWE1-C4341S. After selection with 1 µg/ml puromycin for 4 days, 90% of the cell population was harvested as the initial time point. For the next 42 days, cells were passaged 19 times in a ratio of 1:10 and pellets of harvested cells were frozen for subsequent genomic DNA extraction with Quick-DNA Midiprep Plus Kit (Zymo). A total of 80 µg genomic DNA per sample and time point were used to amplify the sgRNA sequences incorporated into the genome using the NEBNext® Ultra™ II Q5® Master Mix and unique pairs i5 and i7 primers for NGS library index PCR with 28 cycles. PCR products were selected in size and purified from an agarose gel. The DNA content of each library was determined using the Quant-iT™ PicoGreen™ dsDNA Assay-Kit (Thermo) prior to pooling libraries in equimolar ratios for each experiment.

## Next-generation sequencing and data analysis

Pooled libraries were sequenced on the Illumina NovaSeq 6000 platform using single-end sequencing. Demultiplexing of the sequencing reads was performed with Illumina bcl2fastq (version 2.20) and adapters were trimmed with Skewer (version 0.2.2). Reads were mapped to the human genome assembly hg19 using STAR (version 2.5.4)[89] and differential gene expression analysis was conducted using EdgeR (version 3.26.8)[90]. Analysis of ChIPseq, Cut&Run and DSBCapture derived data was performed using Homer (version 4.10.3). sgRNA screen data were analyzed using MAGECK software (version 0.5.9)[91].

## Statistics and reproducibility

All experiments were replicated as indicated in the respective figure legends. Sample sizes were chosen based on previous publications in the field. No statistical method was used to predetermine the sample size. No data was excluded from analysis, except for the analysis of immunofluorescence images of proximity ligation assays. During these analyses, overlapping nuclei and nuclei at the image borders were excluded since these events do not represent an entire single cell. Replicate measurements were taken from distinct samples. Non-parametric statistical tests were chosen when the assumptions of equal

variances and normality were violated. Adjusted p values were derived from post-hoc tests after correction for multiple comparisons, as indicated in the respective figure legends. The Investigators were not blinded during group allocation and outcome assessment.

### Reporting summary

Further information on research design is available in the Nature Portfolio Reporting Summary linked to this article.

## Data availability

All data and reagents are available from the corresponding author upon request. Source data generated in this study are provided in the Supplementary Information and the Source Data files. Commercial Kits and reagents are listed in Supplementary Data 1. The NGS data generated in this study have been deposited in NCBI's Gene Expression Omnibus[92] and are accessible through GEO Series accession number GSE218240. The mass spectrometry data generated in this study have been deposited to the ProteomeXchange Consortium via the PRIDE partner repository[93] with the dataset identifier PXD037677, PXD037680 and PXD037812. Mass spectra were searched against UniProt reference proteome databases for *Mus musculus* (release_2020-10-07) or *Homo sapiens* (release_2019-12-11) (https://www.uniprot.org/) with the Andromeda search engine. Source data are provided with this paper.

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

## Acknowledgements
The authors thank Ksenia Popova for technical assistance; Katharina Zittlau for support with mass spectrometry experiments; Stefanie Schmid and Martin Eilers for sharing plasmids and unpublished data. This work was supported by grant PO1458/7-1 (FOR2314 "Targeting therapeutic windows in essential cellular processes for tumor therapy") from the German Research Foundation (Deutsche Forschungsgemeinschaft) and grant No. 2022.059.1 from the Sander Stiftung to N.P. The authors acknowledge support from the Open Access Publication Fund of the University of Tübingen.

## Author contributions
N.P. conceived the study and designed experiments. E.E., C.J., V.A., and N.P. performed experiments. N.P. and B.M. supervised experiments. N.P. and E.E. wrote the manuscript.

## Funding

## Competing interests
The authors declare no competing interests.
