## [Peer Review File · Nature Communications]

REVIEWER COMMENTS

Reviewer #1 (Remarks to the Author):

In the manuscript entitled “RNAPII-dependent ATM signaling at collisions with replication forks”, the authors provide evidence that elongating RNA polymerase II (Pol II) interacts on chromatin with the ubiquitin-ligase HUWE1 and the ATPase WRNIP1. Under conditions of replication stress and fork stalling, the authors provide evidence that the interaction of a Pol II complex encountering a replisome can lead to the transfer of WRNIP1 from the transcription complex onto the approaching replisome, thereby leading to the activation of ATM pathway, which helps to mitigate the conflict and prevent major DNA damage and double-strand breaks to accumulate. Thus, TRC-activated ATM signalling appears to be a beneficial event for cells and promote genome stability, which is an important contribution in this field.

The data are solid and support the major conclusions from the manuscript. I'm generally supportive of publication in Nature Communications, when the authors can address the following few minor comments:

- 1) The authors use Co-IP experiments at several occasions to show an accumulation of replisome proteins MCM2 and MCM5 with RNAPII (Fig. 1H, Fig. 3D) or ATM (Fig. 6D). MCM2 and MCM5 are also components of licensed replication origins and therefore this increased interaction may also stem from increased collisions between Pol II and MCM double-hexamers licensed in G1 phase. Thus, the authors should repeat these Co-IP experiments in synchronized S-phase cells (e.g by double thymidine block) and/or look at the enrichment of other replisome components that are not part of licensed replication origins (e.g. CDC45, PCNA, DNA polymerase, etc).
- 2) In Figure 2D, the authors show a PLA combination between HUWE1 and WRNIP1 to indicate interaction between the two proteins. As this is a new PLA combination, the authors should provide single antibody controls as well as example images of the stainings. Ideally, the authors could also include a pulse of EdU incorporation to check whether the interaction frequency is changed in S-phase versus non-S-phase cells.
- 3) Figure 2F: It would be beneficial to show the EdU intensity as violin plots instead of box/whisker plots.
- 4) In Figure 4C-D, the authors perform a DSBcapture assay to check the level of DSBs under the conditions of WRNIP1 knockdown and/or HUWE1 mutation. Interestingly, the example gene shows an accumulation of DSB at the promoter of the ZNF555 gene, although the ChIP-Seq data of Pol II seems to indicate that the accumulation of conflicts happens preferentially at the 3' end of genes (Figure 3F). The authors should provide more information on the genomic distribution of the detected DSBs. Can they be more frequently found at promoter, gene body, terminations sites or intergenic regions?
- 5) The effects shown in Figure 5D/E are rather small and no quantification from biological replicates is shown to provide evidence for the reproducibility of these effects.

Reviewer #2 (Remarks to the Author):

In this manuscript, Einig et al propose that HUWE-1 facilitates the interaction between WRNIP1 and elongating RNAPII and HUWE-1 allow the activation of ATM at TRCs in order to restart stalled replisomes. In the absence of this mechanism, genotoxic DSBs ensue. The manuscript presents data that globally support the authors' hypothesis. However, certain points could be strengthened by additional data.

1. Sequential immunoprecipitations should be performed (RNAPII, WRNIP1, HUWE1 and replisome protein, WRNIP1 and HUWE1) in the presence and absence of TRCs to validate the conclusions of the model.
2. The authors concluded from the ChIP-seq expt in Fig 3A that WRNIP1 binds chromatin in a HUWE1-dependent manner. A statistical test should be performed (box plots, violin plots) to show that the difference is significant. The same quantification should be performed for the cut&run expt in Suppl Fig 3A.
3. Re-ChIPs should be performed to validate the conclusions of the model.
4. From the data shown in figure 3, are the genes most associated with WRNIP1 the most highly transcribed genes? It would be useful to see the distribution of WRNIP1 ChIP-seq signal as a function of RNAPII signal. Do the most highly transcribed genes in WT conditions respond differently to TRCs compared to the least transcribed genes, in terms of dependence on WRNIP1 and HUWE1?
5. In figure 4, what is the transcriptional status of the subset of genes shown in panel D ?

Minor point:

The legend to figure 3 refers to supplementary figure 3G, which appears to be an error. Also, there is no description of panel G that I could find.

Reviewer #3 (Remarks to the Author):

The manuscript "RNAPII-dependent ATM signalling at collisions with replication forks" by Dr Popov and colleagues attempts to examining the roles of WRNIP1 and HUWE1 in protecting against transcription-replication conflicts (TRCs) and how TRC activate ATM signalling. In its current form this manuscript, and data presented within, are very difficult to appraise and the reader is asked to work very hard to decipher the implications. PLA and westerns blots examining TRC and DNA damage response respectively, do not provide any evidence of stress at the fork or replicative stress. Moreover, in some cases protein bands (westerns) are difficult to see and appropriate loading controls are missing (see below). Together these issues make it difficult to agree with the authors conclusions, which are frequently unsubstantiated. At a minimum I suggest the manuscript needs re-writing in line with my comments below.

Major concerns:

1. This manuscript relies heavily on PLA between cellular machineries responsible for gene expression and genome duplication to demonstrate TRC. PLA against PCNA and RNAPII, for example, is not a functional assay and only IMPLIES any conflicts. Accompanying westerns examining the DNA damage response provide additional evidence however ultimately these only show the presence of a DNA damage response and not the cause. Replication/replicative stress (or stress at the fork) is defined as: ...'the slowing or stalling of replication fork progression and/or DNA synthesis.' (PMID: 24818779). Additional DNA fibre experiments would validate their results, permitting the conclusions and thus language used.

a. In the abstract on lines 23 and 24 the authors write: 'We show the ATPase WRNIP1 that binds and protects stalled replication forks under stress, associates with RNAPII and limits ATM activation during unperturbed cell cycle.' Firstly, this sentence is confusing and secondly, the authors do not show any evidence that WRNIP1 that binds and protects STALLED replication forks. Similarly, throughout the manuscript the authors refer to stalled forks or replicative stress without providing any evidence. On lines 182-184 the authors write: 'Depletion of WRNIP1 did not increase pATR levels, indicating that ATM signalling is not caused by breakage of stalled DNA replication forks.' Here the authors are implying that WRNIP1 depletion cause replication fork stalling however no evidence has been provided.

b. On lines 146-148 the authors write: 'Indeed, mutation of HUWE1 and depletion of WRNIP1 stimulated binding of the other protein to replication forks (Supplementary Fig. 2G, H).' PLA experiments shown in Supplementary Fig. 2G and H provide no evidence of 'stimulating binding' or presence at the 'replication fork'. Since WRNIP1 interacts with RNAPII (Fig 3B) and TRC are increase in HUWE1-CS cells compared to wild type (Fig 2I) could the increase is WRNIP1-MCM2 foci (Supp Fig 2H) be explained by WRNIP1-RNAPII interaction?

c. On lines 172-173 the authors write: 'We concluded that HUWE1 promotes WRNIP1 association with pS2-RNAPII, which suppresses TRCs at transcription end sites.' The authors provide no data to support this conclusion or that any TRCs occur at transcription end sites.

d. On lines 259-261 the authors write: 'Inhibition of HUWE1 induces dissociation of WRNIP1 from RNAPII, collisions with the replisome and localized ATM activation on RNAPII. This mechanism is evoked under replicative stress to activate ATM and promote DNA repair.' Again, no evidence of replicative stress is provided!

2. In some of the western blots it is difficult to see protein bands and difference between treatments. Moreover, in many case the choice of loading control is inappropriate, inconsistent, or absent. Subsequently, it's difficult to assess the implications of the experiments shown in the blots.

a. Supplementary 2F, Fig 4A, Fig 4E, Supplementary 4B, and Fig 6C: The correct loading for γ H2AX is H2AX especially since HUWE1 mediates H2AX ubiquitination.

b. Supplementary 3C, 4B and 6H: The correct loading for pS2-RNAPII is total RNAPII. Have the authors demonstrated whether total RNAPII levels are the same between experiments/treatments/conditions?

c. Likewise TOTAL protein levels should be assayed for the following pATR, pKAP1, pRPA2, pCHK2.

d. No loading controls are present for Supplementary Fig 6A, 6G and 6H.

e. In some of the immunoprecipitation experiment the protein bands presented are hardly discernible.

3. On lines 138-140 the authors write: 'Depletion of WRNIP1 diminished EdU incorporation in HUWE1-WT but not in HUWE1-CS cells (Fig. 2F; Supplementary Fig. 2F), suggesting that HUWE1 and WRNIP1 regulate DNA replication via a common mechanism.' Reduced EdU incorporation in mutant HUWE1-CS cells or WRNIP1-depleted cells provides no mechanistic insight and the sentence should be modified accordingly.

4. On lines 193-195 the authors write: 'The double mutant cells also showed a strong increase in pS4/8-RPA2 (Fig. 4A), a marker of collapsed replication forks and a target of DNA-PK, suggesting that HUWE1 and WRNIP1 can redundantly prevent fork degradation.' It is true increased pS4/8-RPA2 is observed following replication fork collapse. However, DNA-PK is activated upon DSB and pS4/8-RPA2 is also induced by UV light (PMID: 9139719). Here the authors must modify their language.

5. What is the concentration and treatment duration of hydroxyurea used in EACH experiment? In Fig 6F a concentration of 250 μ M hydroxyurea (HU) is used for 4 days. Is this the treatment conditions for ALL HU experiments? As the author will be aware the concentration and duration of HU treatment dictates the type of stress and interpretations. Thus, it is difficult to appraise the experimental outcome in all experiments using HU! Furthermore, pKAP1 and pCHK2 westerns are insufficient to demonstrate ATM signalling following HU treatment, as suggested on line 320 of the discussion.

6. On lines 193-195 the authors write: 'Like TRCs, ATM signalling in HUWE1-CS cells was diminished by treatment with CDK9 inhibitor and AZD4573 or by triptolide (Fig. 5A), suggesting that elongating RNAPII promotes ATM activation and leading us to analyze the underlying mechanisms.' Both ATM or ATR can phosphorylate KAP1 and Chk2 (PMID: 16741947). Thus, to show the effects of transcriptional inhibition of ATM signalling, pATM (p1981) foci or more specific ATM targets are need with and without transcriptional inhibition. The manuscript would also benefit from some consistency when examining the DNA damage response in multiple experiments. For example Fig 4A vs 4E vs 5A.

7. The NGS data requires further data mining and explanation. The DSB capture suggests DSB occur around gene TSS, however their location is not discussed in the text. How does this data (DSBCapture) correspond with the WRN1-ChIP-seq and pS2-RNAPII-ChIP-seq data? On lines 172-173 the authors write: 'We concluded that HUWE1 promotes WRNIP1 association with pS2-RNAPII, which suppresses TRCs at transcription end sites.' Do DSB in HUWE1-CS cells localised at transcription end sites?

8. Fig 5F identifies RAD50 in a sgRNA library screen for viability of HUWE1-CS cells. The relevance and importance are not mentioned or followed up. This is another example of the authors expecting the reader to read between the lines rather than explicitly emphasising the importance. This finding could have been confirmed by shRNA/siRNA knockdown of RAD50, NSB1 or MRE11. The authors allude to the activation of ATM signalling via RNAPII-MRN in the discussion. Presumably RAD50 knockdown would inhibit ATM-signalling and reduce cell viability.

9. On lines 290-294 the authors write: 'The translocation of WRNIP1 would likely be most efficient at RNAPII collisions with the replisome (TRCs), which are induced by HUWE1 mutation. Based on previous studies, we propose that WRNIP1 stabilizes replication forks that collide with RNAPII. Supporting this model, depletion of WRNIP1 in HUWE1-CS cells reverts the increase in TRC levels.' In light of the decreased pKAP1 and pH2AX shown in Supp Fig 4B upon WRNIP1 overexpression, surely over expression of HA-WRNIP1 in HUWE1-CS cells would provide better evidence for this conclusion??

10. The following statements in the discussion require more context and better arguments to support these claims:

a. Line 304- 306: TRCs thus appear to provide a controlled mechanism for stalling of replication forks that facilitates fork stabilization and restart during recovery from stress.

b. Line 315-317: 'One mechanism that triggers ATM recruitment can involve translocation of WRNIP1, for example, if it sterically interferes with ATM binding to the MRN complex.' Is this statement based on observations in this manuscript or others?

Minor:

1. In this study HUWE1 was mutated using CRISPR to generate a catalytic dead version (HUWE1-CS). The authors must explicitly describe if these cells are homo or heterozygous mutants, presumably the former. Given that these cells display reduced viability and increased DSB, a rescue experiment with wild-type HUWE1 would alleviate any concerns these cells acquire genomic instability issues

2. Throughout the manuscript HCT116 cells have been used for NGS and mass spec experiments. Then in Fig 5B MOUSE EMBRYONIC FIBROBLASTS are used for a pS2-RNAPII reactome. Is this data need? Especially since a study by Salifou et al., 2021 already show MRN globally localises with RNAPII!

3. Similarly, Fig 5F identifies RAD50 in a sgRNA library screen for viability of HUWE1-CS cells. The relevance and importance are not mentioned or followed up. This is another example of the authors expecting the reader to read between the lines rather than explicitly emphasising the importance. This finding could have be confirmed by shRNA/siRNA knockdown.

4. The data in Fig 3C would benefit from Supp Fig 3C being discussed first.

5. No figure legend for Fig 3G.

6. No statistics are provided for Supp Fig 4A and Fig 6A.

7. On lines 'The transfer of WRNIP1 from RNAPII onto replication forks is evoked upon global fork stalling (e.g., under hydroxyurea treatment) but is also likely to occur locally and transiently during normal cell cycle to coordinate DNA replication with RNAPII transcription (MacAlpine et al., 2004, Liu et al., 2021, Saponaro, 2022).' The inclusion of these reference at the end of this sentence is confusing, since they do not refer to WRNIP1 in any context. Also the word 'transfer' must be changed as it suggests WRNIP1 moves directly from RNAPII to the replication forks.

8. A complete list of antibodies are not provided in the reporting summary.

Point-by-point reply to reviewer comments

Reviewer 1

In the manuscript entitled “RNAPII-dependent ATM signaling at collisions with replication forks”, the authors provide evidence that elongating RNA polymerase II (Pol II) interacts on chromatin with the ubiquitin-ligase HUWE1 and the ATPase WRNIP1. Under conditions of replication stress and fork stalling, the authors provide evidence that the interaction of a Pol II complex encountering a replisome can lead to the transfer of WRNIP1 from the transcription complex onto the approaching replisome, thereby leading to the activation of ATM pathway, which helps to mitigate the conflict and prevent major DNA damage and double-strand breaks to accumulate. Thus, TRC-activated ATM signalling appears to be a beneficial event for cells and promote genome stability, which is an important contribution in this field.

We thank the reviewer for the appreciation of our work and suggestions for experiments.

The data are solid and support the major conclusions from the manuscript. I'm generally supportive of publication in Nature Communications, when the authors can address the following few minor comments:

1. The authors use Co-IP experiments at several occasions to show an accumulation of replisome proteins MCM2 and MCM5 with RNAPII (Fig. 1H, Fig. 3D) or ATM (Fig. 6D). MCM2 and MCM5 are also components of licensed replication origins and therefore this increased interaction may also stem from increased collisions between Pol II and MCM double-hexamers licensed in G1 phase. Thus, the authors should repeat these Co-IP experiments in synchronized S-phase cells (e.g by double thymidine block) and/or look at the enrichment of other replisome components that are not part of licensed replication origins (e.g. CDC45, PCNA, DNA polymerase, etc).

Since licensed origins that do not fire remain associated with chromatin until the G2/M phase, we followed the second suggestion. We performed immunoblotting of the RNAPII immunoprecipitates with PCNA antibodies and found an increased association in HUWE1-CS cells (Fig. 1G). Likewise, immunoprecipitation with CDC45 and PCNA antibodies yielded higher levels of RNAPII in HUWE1-CS cells (Supplementary Fig. 1H,I), consistent with the idea that mutation of HUWE1 promotes RNAPII collisions with active replication forks.

2. In Figure 2D, the authors show a PLA combination between HUWE1 and WRNIP1 to indicate interaction between the two proteins. As this is a new PLA combination, the authors should provide single antibody controls as well as example images of the stainings. Ideally, the authors could also include a pulse of EdU incorporation to check whether the interaction frequency is changed in S-phase versus non-S-phase cells.

We performed PLA assays with HUWE1-WRNIP1 antibodies after pulse-labeling cells with EdU and included single antibody controls. EdU positive cells have significantly more proximity pairs, indicating that HUWE1-WRNIP1 interaction is more pronounced in the S-phase. These results are shown in Fig. 2D and described on page 4.

3. *Figure 2F: It would be beneficial to show the EdU intensity as violin plots instead of box/whisker plots.*

We provide the violin plot for EdU intensity in Fig. 2H.

4. *In Figure 4C-D, the authors perform a DSBcapture assay to check the level of DSBs under the conditions of WRNIP1 knockdown and/or HUWE1 mutation. Interestingly, the example gene shows an accumulation of DSB at the promoter of the ZNF555 gene, although the ChIP-Seq data of Pol II seems to indicate that the accumulation of conflicts happens preferentially at the 3' end of genes (Figure 3F). The authors should provide more information on the genomic distribution of the detected DSBs. Can they be more frequently found at promoter, gene body, terminations sites or intergenic regions?*

Interestingly, DSBs in WRNIP1-depleted HUWE1-CS cells occur predominantly at promoter regions and around 25% of DSBs localize to gene bodies and intergenic regions. We provide an example of an intergenic peak at the region which follows a transcription end site and a graph with annotation of the DSBs in the double mutant cells (Supplementary Fig. 4C, D) and describe these data on page 6 .

Most of the break sites appear to be spatially distinct from regions, where RNAPII accumulates in single mutant cells (shWRNIP1-HUWE1-WT and shCtrl-HUWE1-CS), which we propose to represent the TRC sites. One possible interpretation of this result is that in the absence of both functional HUWE1 and WRNIP1, forks cannot stall at these sites but progress further and collapse in the vicinity of promoters, which are most prone to breakage.

5) *The effects shown in Figure 5D/E are rather small and no quantification from biological replicates is shown to provide evidence for the reproducibility of these effects.*

We repeated these assays and now show stronger experiments with quantification of three biological replicates in Fig. 5E and Fig. 5F.

Reviewer #2 (Remarks to the Author):

In this manuscript, Einig et al propose that HUWE-1 facilitates the interaction between WRNIP1 and elongating RNAPII and HUWE-1 allow the activation of ATM at TRCs in order to restart stalled

replisomes. in the absence of this mechanism, genotoxic DSBs ensue. The manuscript presents data that globally support the authors' hypothesis. However, certain points could be strengthened by additional data.

We thank the reviewer for the insightful comments and suggestions.

1. Sequential immunoprecipitations should be performed (RNAPII, WRNIP1, HUWE1 and replisome protein, WRNIP1 and HUWE1) in the presence and absence of TRCs to validate the conclusions of the model.

We immunoprecipitated HUWE1 from crosslinked HUWE1-WT and HUWE1-CS cells followed by elution of the antibody-bound complexes with SDS, second immunoprecipitation with the pS2-RNAPII antibody and immunoblotting. We were able to detect WRNIP1 in these precipitates, suggesting that HUWE1 and WRNIP1 associate with RNAPII as a complex (Supplementary Fig. 3C).

In sequential immunoprecipitation with WRNIP1 followed by the pS2-RNAPII antibodies, we were unable to detect replisome proteins in either HUWE1-WT or HUWE1-CS cells (Supplementary Fig. 2F). This result is consistent with the data obtained in the PLA assays with WRNIP1-MCM2 and WRNIP1-RNAPII antibodies that indicate preferential binding of WRNIP1 to RNAPII in the low TRC conditions (HUWE1-WT cells) and to the replisome in high-TRC conditions (HUWE1-CS cells or hydroxyurea treatment) (Supplementary Fig. 2; Fig. 3D and Fig. 6A). Also, since straight immunoprecipitation of RNAPII readily recovers replisome proteins in HUWE1-CS cells, this result indicates that WRNIP1-bound RNAPII complexes are not engaged in TRCs, consistent with the idea that WRNIP1 promotes TRC resolution. These experiments are described on page 5 of the results section.

2. The authors concluded from the ChIP-seq expt in Fig 3A that WRNIP1 binds chromatin in a HUWE1-dependent manner. A statistical test should be performed (box plots, violin plots) to show that the difference is significant. The same quantification should be performed for the cut&run expt in Suppl Fig 3A.

We quantified WRNIP1 ChIP-seq and Cut&Run data for genes with strongest Wrnip1 enrichment and found a significant difference between HUWE1-WT and HUWE1-CS cells. These data are shown in Fig. 3B and Supplementary Fig.3B

3. Re-ChIPs should be performed to validate the conclusions of the model.

We performed WRNIP1 ChIP followed by elution and RNAPII ChIP and found a strongly reduced binding at several tested transcription start sites (Fig. 3E), in line with PLA assays (Fig. 3D) and WRNIP1 ChIP/Cut&Run experiments (Fig. 3A; Supplementary Fig. 3A). We describe this experiment on page 4.

4. From the data shown in figure 3, are the genes most associated with WRNIP1 the most highly transcribed genes? it would be useful to see the distribution of WRNIP1 ChIP-seq signal as a function of RNAPII signal. Do the most highly transcribed genes in WT conditions respond differently to TRCs compared to the least transcribed genes, in terms of dependence on WRNIP1 and HUWE1?

We analyzed the relationship between WRNIP1 binding and total or pS2-RNAPII binding (normalized tags) and found a strong correlation (Supplementary Fig. 3G).

We also analyzed the expression levels of highly and low expressed genes in shCtrl-HUWE1-WT vs shWRNIP1-HUWE1-WT and shCtrl-HUWE1-CS (high incidence of TRCs) and shWRNIP1-HUWE1-CS cells (low TRCs) and found no strong differences in expression levels (Please see Fig. 1 below).

Figure 1: RNAseq tag counts for the 10% of highest or lowest expressed genes in WT shCtrl conditions.

5. In figure 4, what is the transcriptional status of the subset of genes shown in panel D ?

The genes with DSB peaks are expressed slightly higher (have a modestly higher number of normalized RNA-seq tags) compared to all genes (Supplementary Fig. 4F).

On average, genes with DSBs are not significantly deregulated in either of the four cell lines, suggesting that the sensitivity to breakage of a given gene is largely uncoupled from its transcriptional output (Supplementary Fig. 4G). We described this analysis on page 6.

Minor point:

The legend to figure 3 refers to supplementary figure 3G, which appears to be an error. Also, there is no description of panel G that I could find.

We added the description of Figure 3G (now Supplementary Fig. 3J).

Reviewer #3 (Remarks to the Author):

The manuscript "RNAPII-dependent ATM signalling at collisions with replication forks" by Dr Popov and colleagues attempts to examining the roles of WRNIP1 and HUWE1 in protecting against transcription-replication conflicts (TRCs) and how TRC activate ATM signaling. In its current form this manuscript, and data presented within, are very difficult to appraise and the reader is asked to work very hard to decipher the implications. PLA and westerns blots examining TRC and DNA damage response respectively, do not provide any evidence of stress at the fork or replicative stress. Moreover, in some cases protein bands (westerns) are difficult to see and appropriate loading controls are missing (see below). Together these issues make it difficult to agree with the authors conclusions, which are frequently unsubstantiated. At a minimum I suggest the manuscript needs re-writing in line with my comments below.

We thank the reviewer for the constructive criticism and suggestions, which helped us strengthen the manuscript. We performed new experiments that document reduced fork progression in WRNIP1-depleted and HUWE1-CS cells (Fig. 2F). We repeated several immunoprecipitation and immunoblotting experiments, as specified in the comments below and provide quantification for the key experiments (Fig. 1G; Fig. 5E,F). We also adjusted the phrasing according to the following suggestions to clarify the logic of the experiments and interpretation of the results.

Major concerns:

1. *This manuscript relies heavily on PLA between cellular machineries responsible for gene expression and genome duplication to demonstrate TRC. PLA against PCNA and RNAPII, for example, is not a functional assay and only IMPLIES any conflicts. Accompanying westerns examining the DNA damage response provide additional evidence however ultimately these only show the presence of a DNA damage response and not the cause. Replication/replicative stress (or stress at the fork) is defined as: ...'the slowing or stalling of replication fork progression and/or DNA synthesis.' (PMID: 24818779). Additional DNA fibre experiments would validate their results, permitting the conclusions and thus language used.*

To the best of our knowledge, PLA is essentially the only technique to visualize TRCs and the RNAPII / PCNA antibody pair is by far the most common combination. To strengthen our conclusions, we performed immunoprecipitation from the crosslinked cells with RNAPII antibodies, followed by immunoblotting for replisome proteins (Fig. 1G; Fig. 3F). We also performed immunoprecipitations with PCNA and CDC45 antibodies, followed by immunoblotting with the RNAPII antibodies, which provide further support for the results of the PLA assay (Supplementary Fig. 1H,I).

We performed new experiments that document slower replication fork progression in WRNIP1-depleted cells (Fig. 2F), which is a bona fide characteristic of replicative stress, as highlighted by the reviewer. The DNA fiber data for HUWE1-CS cells are shown in Figure 1D and are consistent with previous observations (Choe et al, 2016). The Edu incorporation experiments (Fig. 2G) show decreased incorporation in HUWE1-CS or shWRNIP1 cells, compared to shCtrl-

HUWE1-WT cells supporting the conclusion that deficiency in either protein interferes with DNA replication.

a. *In the abstract on lines 23 and 24 the authors write: 'We show the ATPase WRNIP1 that binds and protects stalled replication forks under stress, associates with RNAPII and limits ATM activation during unperturbed cell cycle.' Firstly, this sentence is confusing and secondly, the authors do not show any evidence that WRNIP1 that binds and protects STALLED replication forks. Similarly, throughout the manuscript the authors refer to stalled forks or replicative stress without providing any evidence. On lines 182-184 the authors write: 'Depletion of WRNIP1 did not increase pATR levels, indicating that ATM signalling is not caused by breakage of stalled DNA replication forks.' Here the authors are implying that WRNIP1 depletion cause replication fork stalling however no evidence has been provided.*

In the abstract we referred to the published data that WRNIP1 interacts with and protects stalled replication forks (Kanu et al., 2016, Leuzzi et al., 2016). We have now rephrased this sentence for clarity.

Since WRNIP1 association with replisome proteins increases under conditions of replicative stress, induced by HUWE1 mutation or hydroxyurea treatment (Supplementary Fig. 2H,I; Fig. 6A), we conclude that WRNIP1 preferentially associates with stalled forks. Furthermore, using antibodies to Ub-PCNA, a well-established marker of stalled forks, we were able to immunoprecipitate WRNIP1 from lysates of HU-treated HCT116 cells (please see Figure 2 below). We adjusted the phrasing throughout the manuscript to more explicitly refer to the published observations.

In the original manuscript we showed that HUWE1 mutation reduces replication fork progression using fiber assay (Fig. 1D), demonstrating HUWE1-CS cells experience replicative stress. This result is consistent with previous studies showing a reduced fork progression upon HUWE1 depletion (Choe et al., 2016). We now used DNA fiber assay to show that depletion of WRNIP1 slows down replication forks (Fig. 2F), demonstrating that WRNIP1 deficiency also induces replicative stress.

Figure 2: Immunoprecipitation of ub-PCNA after treatment with 1 mM HU for 5 h followed by immunoblot analysis.

b. *On lines 146-148 the authors write: 'Indeed, mutation of HUWE1 and depletion of WRNIP1 stimulated binding of the other protein to replication forks (Supplementary Fig. 2G, H).' PLA experiments shown in Supplementary Fig. 2G and H provide no evidence of 'stimulating binding' or presence at the 'replication fork'. Since WRNIP1 interacts with RNAPII (Fig 3B) and TRC are*

increase in HUWE1-CS cells compared to wild type (Fig 2I) could the increase in WRNIP1-MCM2 foci (Supp Fig 2H) be explained by WRNIP1-RNAPII interaction?

As done by many researchers for other antibodies (e.g., PCNA - RNAPII), we interpret the increased number of proximity pairs in the PLA assay with WRNIP1 and MCM2 antibodies as an increase in the number of binding events, and therefore propose that more PLA foci implies stronger interaction (Supplementary Fig. 2H). To support this conclusion, we immunoprecipitated WRNIP1 from crosslinked cells followed by immunoblotting and found an increased signal for MCM2, PCNA and CDC45 (Supplementary Fig. 2I).

Our data suggest that WRNIP1 binding to RNAPII and to replication forks is mutually exclusive. For example, in HUWE1-WT cells, WRNIP1 preferentially associates with RNAPII (Fig. 3B). Mutation of HUWE1 increases TRCs and diminishes the WRNIP1-RNAPII interaction (Fig. 3D), but increases WRNIP1 binding to MCM2, MCM5, CDC45 and PCNA (Supplementary Fig. 2H,I). Treatment with HU also induces TRCs and diminishes WRNIP1 binding to RNAPII but promotes WRNIP1 association with MCM2 (Fig. 6A). Therefore, the increase in WRNIP1-MCM2 PLA foci cannot be explained by WRNIP1-RNAPII interaction.

c. On lines 172-173 the authors write: 'We concluded that HUWE1 promotes WRNIP1 association with pS2-RNAPII, which suppresses TRCs at transcription end sites.' The authors provide no data to support this conclusion or that any TRCs occur at transcription end sites.

We observe accumulation of pS2-RNAPII at gene bodies and transcription end sites in shWRNIP1 cells and HUWE1-CS cells relative to shCtrl HUWE1-WT, whereas in shWRNIP1-HUWE1-CS cells this accumulation is rescued (Figure 3G). Since TRCs also increase in single mutant cells and decrease in the double mutant cells (Figure 2K) (based on PLA and immunoprecipitation data), we hypothesize that the regions where RNAPII accumulates may represent putative TRC sites. We agree that this idea is largely based on correlations and adjusted the description of these data in the manuscript (page 5).

d. On lines 259-261 the authors write: 'Inhibition of HUWE1 induces dissociation of WRNIP1 from RNAPII, collisions with the replisome and localized ATM activation on RNAPII. This mechanism is evoked under replicative stress to activate ATM and promote DNA repair.' Again, no evidence of replicative stress is provided!

In this sentence we refer to HU treatment, which depletes the cellular pool of dNTPs and stalls replication forks and is therefore used by many laboratories as a prime example of conditions to induce replicative stress (for example, Choe et al., 2016, Crosetto et al., 2008). We now performed the fiber assay under HU treatment in HCT116 cells to formally show that hydroxyurea induces replicative stress in this cell line (please see Figure 3 below)

Figure 3. DNA fiber assay in HCT116 cells. 1 mM HU or DMSO was added simultaneously with CldU. A) Representative images of DNA tracks. B) Quantification of fiber length. Significance was determined by Kruskal-Wallis test followed by Dunn's multiple comparison. **** $p \leq 0.0001$, ns: not significant.

2. *In some of the western blots it is difficult to see protein bands and difference between treatments. Moreover, in many case the choice of loading control is inappropriate, inconsistent, or absent. Subsequently, it's difficult to assess the implications of the experiments shown in the blots.*

We repeated many of the weaker immunoblotting experiments and provided stronger data with quantification as detailed below.

a. *Supplementary 2F, Fig 4A, Fig 4E, Supplementary 4B, and Fig 6C: The correct loading for γ H2AX is H2AX especially since HUWE1 mediates H2AX ubiquitination.*

We added the immunoblots for total H2AX in Fig. 4A, E; Supplementary Fig. 2G, 4B; Fig. 5A; Fig. 6C.

b. *Supplementary 3C, 4B and 6H: The correct loading for pS2-RNAPII is total RNAPII. Have the authors demonstrated whether total RNAPII levels are the same between experiments/treatments/conditions?*

We determined the levels of total RNAPII in these experiments and included these data to Supplementary Fig. 3C (now Supplementary Fig. 3E). We did not make any statement regarding pS2-RNAPII in Supplementary Fig. 4B and Supplementary Fig. 6H and therefore removed the pS2-RNAPII panels from these figures.

c. *Likewise TOTAL protein levels should be assayed for the following pATR, pKAP1, pRPA2, pCHK2.*

We include the immunoblots that show the total levels of ATR, KAP1, RPA2 and CHK2 in Fig. 4A, E, Supplementary Fig. 4B, Fig. 6C.

d. *No loading controls are present for Supplementary Fig 6A, 6G and 6H.*

We now included the loading controls for the experiments shown in Supplementary Fig. 6A, H and Supplementary Fig. 6G (now Supplementary Fig. 6I).

e. *In some of the immunoprecipitation experiment the protein bands presented are hardly discernible.*

We repeated the experiments shown in Figures 5E and 5F and provide stronger data with quantification of three biological replicates.

3. *On lines 138-140 the authors write: 'Depletion of WRNIP1 diminished EdU incorporation in HUWE1-WT but not in HUWE1-CS cells (Fig. 2F; Supplementary Fig. 2F), suggesting that HUWE1 and WRNIP1 regulate DNA replication via a common mechanism.' Reduced EdU incorporation in mutant HUWE1-CS cells or WRNIP1-depleted cells provides no mechanistic insight and the sentence should be modified accordingly.*

We agree that this experiment does not provide evidence of the underlying mechanism and adjusted the phrasing on page 4.

4 *On lines 193-195 the authors write: 'The double mutant cells also showed a strong increase in pS4/8-RPA2 (Fig. 4A), a marker of collapsed replication forks and a target of DNA-PK, suggesting that HUWE1 and WRNIP1 can redundantly prevent fork degradation.' It is true increased pS4/8-RPA2 is observed following replication fork collapse. However, DNA-PK is activated upon DSB and pS4/8-RPA2 is also induced by UV light (PMID: 9139719). Here the authors must modify their language.*

We adjusted the description of the data to acknowledge that DNA-PK is one candidate kinase (page 6).

5. *What is the concentration and treatment duration of hydroxyurea used in EACH experiment? In Fig 6F a concentration of 250 μ M hydroxyurea (HU) is used for 4 days. Is this the treatment conditions for ALL HU experiments? As the author will be aware the concentration and duration of HU treatment dictates the type of stress and interpretations. Thus, it is difficult to appraise the experimental outcome in all experiments using HU! Furthermore, pKAP1 and pCHK2*

westerns are insufficient to demonstrate ATM signalling following HU treatment, as suggested on line 320 of the discussion.

We now included the concentration and duration of hydroxyurea (HU) treatment and other treatments in the figure legend for all experiments. In most experiments we used a standard concentration of 1 mM HU for the indicated time. In the experiments mentioned in this comment (now shown in Fig. 6G and Supplementary Fig. 6J), we used a lower dose for a longer time to avoid rapid cell death.

Activation of ATM in response to HU has been documented by previous studies (for example, Kanu et al., 2016). We show additionally that another substrate of ATM, pS25-53BP1 also increases following HU treatment in a RAD50-dependent manner, which we document by immunoblots shown in Supplementary Figure 6F. Please also see answers to comments below for additional evidence of ATM activation upon mutation of HUWE1 or depletion of WRNIP1.

6. *On lines 193-195 the authors write: 'Like TRCs, ATM signalling in HUWE1-CS cells was diminished by treatment with CDK9 inhibitor and AZD4573 or by triptolide (Fig. 5A), suggesting that elongating RNAPII promotes ATM activation and leading us to analyze the underlying mechanisms.' Both ATM or ATR can phosphorylate KAP1 and Chk2 (PMID: 16741947). Thus, to show the effects of transcriptional inhibition of ATM signalling, pATM (p1981) foci or more specific ATM targets are need with and without transcriptional inhibition. The manuscript would also benefit from some consistency when examining the DNA damage response in multiple experiments. For example Fig 4A vs 4E vs 5A.*

To corroborate our conclusion on ATM activation, we assessed levels of pATM and/or pS25-53BP1 in the four analyzed cell lines (Supplementary Fig. 4A) and in HUWE1-WT and HUWE1-CS cells treated with triptolide or control (Supplementary Figure 5A,B). We also added control panels corresponding to total proteins to all the experiments analyzing activation of ATM signaling (Fig. 4A,E, 5A, 6C, Supplementary Fig. 4B).

7. *The NGS data requires further data mining and explanation. The DSB capture suggests DSB occur around gene TSS, however their location is not discussed in the text. How does this data (DSBCapture) correspond with the WRN1-ChIP-seq and pS2-RNAPII-ChIP-seq data? On lines 172-173 the authors write: 'We concluded that HUWE1 promotes WRNIP1 association with pS2-RNAPII, which suppresses TRCs at transcription end sites.' Do DSB in HUWE1-CS cells localised at transcription end sites?*

We found that the DSB capture signal (normalized sequencing tags) strongly correlates with WRNIP1 ChIP-seq and pS2-RNAPII ChIP-seq signals - these data are shown in Supplementary Fig. 4E.

Most of DSB peaks are found at promoters and approximately 25% of DSBs localize to intragenic sites, transcription end sites and intergenic regions. We provide an example of the intergenic peak adjacent to a transcription end site and a graph with annotation of the DSBs in the double mutant cells (Supplementary Fig. 4C, D) and describe these data on page 6.

Our CHIP-seq experiments show accumulation of RNAPII in the 3' regions of a group of genes upon mutation of HUWE1 or depletion of WRNIP1. As this accumulation correlates with the number of PCNA-RNAPII in the PLA assay, we hypothesize that these genes represent putative TRC sites. To the best of our knowledge, no assay allows precise mapping of TRC sites, so this certainly is a speculation and we adjusted the phrasing describing this result on page 5.

Since DSBs occur at regions distinct from these sites, one can propose that in the absence of functional HUWE1 and WRNIP1, replication forks do not stably arrest during collisions with RNAPII (shown by low number of foci in RNAPII-PCNA PLA) but progress further and collapse at sites, identified by the DSB capture experiment. Although the exact mechanism of DSB formation in WRNIP1-depleted HUWE1-CS cells remains hypothetical, our data strongly support the model that TRCs promote ATM signaling and DNA repair.

8. *Fig 5F identifies RAD50 in a sgRNA library screen for viability of HUWE1-CS cells. The relevance and importance are not mentioned or followed up. This is another example of the authors expecting the reader to read between the lines rather than explicitly emphasising the importance. This finding could have been confirmed by shRNA/siRNA knockdown of RAD50, NSB1 or MRE11. The authors allude to the activation of ATM signalling via RNAPII-MRN in the discussion. Presumably RAD50 knockdown would inhibit ATM-signalling and reduce cell viability.*

We validated the results of the screen by depleting RAD50 with two independent lentiviral vectors. Depletion of RAD50 compromises survival of HUWE1-CS cells with an opposite effect on HUWE1-WT cells (Fig. 5H). Furthermore, depletion of RAD50 diminishes phosphorylation of KAP1 indicative of compromised ATM signaling in HUWE1-CS cells compared to HUWE1-WT cells (Fig. 5C), abolishes recruitment of ATM to RNAPII (Supplementary Fig. 5F) and blocks HU-induced phosphorylation of ATM targets KAP1 and 53BP1 (Figure 6D, Supplementary Fig. 6F).

9. *On lines 290-294 the authors write: 'The translocation of WRNIP1 would likely be most efficient at RNAPII collisions with the replisome (TRCs), which are induced by HUWE1 mutation. Based on previous studies, we propose that WRNIP1 stabilizes replication forks that collide with RNAPII. Supporting this model, depletion of WRNIP1 in HUWE1-CS cells reverts the increase in TRC levels.' In light of the decreased pKAP1 and pH2AX shown in Supp Fig 4B upon WRNIP1 overexpression, surely over expression of HA-WRNIP1 in HUWE1-CS cells would provide better evidence for this conclusion??*

We analyzed the levels of phosphorylated ATM targets in HUWE1-WT and HUWE1-CS cells, expressing exogenous WRNIP1 protein and found a strong reduction, in line with reviewer's suggestion. These data are shown in Supplementary Fig. 4B and described on page 5.

10. *The following statements in the discussion require more context and better arguments to support these claims:*

a. *Line 304- 306: TRCs thus appear to provide a controlled mechanism for stalling of replication forks that facilitates fork stabilization and restart during recovery from stress.*

This statement is based on the observation that induction of TRCs under many conditions in our experiments is not accompanied by DNA damage (depletion of WRNIP1, HUWE1 mutation, short-term HU treatment). In contrast, inhibition of TRCs under these conditions (e.g., by RNAPII inhibition or combined mutation of WRNIP1 and HUWE1), led to accumulation of DNA damage. For example, when TRCs, induced under hydroxyurea treatment, are diminished by inhibition of RNAPII, the levels of DNA damage increases (Fig. 6B,F).

b. *Line 315-317: 'One mechanism that triggers ATM recruitment can involve translocation of WRNIP1, for example, if it sterically interferes with ATM binding to the MRN complex.' Is this statement based on observations in this manuscript or others?*

This statement is based on our finding that we can detect Mre11 in WRNIP1 immunoprecipitates in unchallenged cells but not HU treatment (Supplementary Fig. 6H). This correlates with the decrease in WRNIP1- RNAPII association and an increase in WRNIP1-MCM2 association under HU (Fig. 6A). Since depletion of WRNIP1 also promotes ATM recruitment to RNAPII, we can hypothesize that dissociation of WRNIP1 from RNAPII upon hydroxyurea treatment can allow ATM binding to RNAPII. We addressed this issue in the discussion on page 9.

Minor:

1. *In this study HUWE1 was mutated using CRISPR to generate a catalytic dead version (HUWE1-CS). The authors must explicitly describe if these cells are homo or heterozygous mutants, presumably the former. Given that these cells display reduced viability and increased DSB, a rescue experiment with wild-type HUWE1 would alleviate any concerns these cells acquire genomic instability issues*

The genotyping PCR data show that our HUWE1-CS and HUWE1-WT cell lines are homozygous. We generated HUWE1-WT cells using the same strategy as HUWE1-CS cells in parallel specifically to control for random genetic events. As we show using comet assays and DSB-capture assays, the HUWE1-CS cells have a low level of DNA breakage, which is completely absent in HUWE1-WT cells, showing that this effect results from the specific mutation rather than the cell line generation strategy. We agree that the rescue experiment would provide a good control, but it is technically very hard to accomplish - HUWE1 is a gigantic protein (cDNA of ca. 16 kb) and it is virtually impossible to stably express the full-length protein for reconstitution experiments.

2. *Throughout the manuscript HCT116 cells have been used for NGS and mass spec experiments. Then in Fig 5B MOUSE EMBRYONIC FIBROBLASTS are used for a pS2-RNAPII*

reactome. Is this data need? Especially since a study by Salifou et al., 2021 already show MRN globally localises with RNAPII!

We used the MEF mass spectrometry data to support the logic underlying our experiments. We agree with the reviewer that the use of MEFs may seem to be somewhat inappropriate considering that HCT116 cells have been used for all other experiments. We therefore show a schematic to illustrate the published connection between RNAPII and the MRN /ATM complex, citing Salifou et al., 2021 and Sharma et al., 2021 in Fig. 5B. The MEF mass spectrometry data are now shown in Supplementary Fig. 5C since they clearly support our model and show that the MRN-RNAPII interaction is not limited to human cell lines or transformed cells.

3. *Similarly, Fig 5F identifies RAD50 in a sgRNA library screen for viability of HUWE1-CS cells. The relevance and importance are not mentioned or followed up. This is another example of the authors expecting the reader to read between the lines rather than explicitly emphasising the importance. This finding could have be confirmed by shRNA/siRNA knockdown.*

We now performed several experiments with cells expressing RAD50 shRNAs. We show that depletion of RAD50: 1) reduces viability of HUWE1-CS but not of HUWE1-WT cells, validating the results of the screen (Fig. 5H); 2) diminishes ATM binding to RNAPII (Supplementary Fig. 5F) and downregulates KAP1 phosphorylation in HUWE1-CS cells (Fig. 5C), indicative of reduced ATM activation; 3) diminishes hydroxyurea-induced phosphorylation of KAP1 and 53BP1 (Fig. 6D, Supplementary Fig. 6F), indicating that activation of ATM upon HU requires RAD50.

4. *The data in Fig 3C would benefit from Supp Fig 3C being discussed first.*

We performed additional experiments and adjusted the description of the results accordingly. In our opinion, the PLA data should be introduced first, as it supports the ChIP and immunoprecipitation assays (Fig. 3A,B,C). The effect on pS2-RNAPII levels prompted us to analyze RNAPII distribution by ChIP-seq, so we consider this order of presentation more logical for our storyline.

5. *No figure legend for Fig 3G.*

We added figure legend for this panel, which is now shown in Supplementary Fig. 3J.

6. *No statistics are provided for Supp Fig 4A and Fig 6A.*

We added statistics to Supplementary Fig. 4A and Fig. 6A.

7. On lines 'The transfer of WRNIP1 from RNAPII onto replication forks is evoked upon global fork stalling (e.g., under hydroxyurea treatment) but is also likely to occur locally and transiently during normal cell cycle to coordinate DNA replication with RNAPII transcription (MacAlpine et al., 2004, Liu et al., 2021, Saponaro, 2022).' The inclusion of these reference at the end of this sentence is confusing, since they do not refer to WRNIP1 in any context. Also the word 'transfer' must be changed as it suggests WRNIP1 moves directly from RNAPII to the replication forks.

We used these references since they address coordination of RNAPII transcription and DNA replication - the process in which the function of WRNIP1 can be instrumental. We added a reference to a study on WRNIP1 for clarity.

We now adjusted the phrasing in the abstract and throughout the manuscript to acknowledge the fact that WRNIP1 may not move directly from RNAPII to the replisome.

8. A complete list of antibodies are not provided in the reporting summary.

We included the complete list in the reporting summary.

References

- CHOE, K. N., NICOLAE, C. M., CONSTANTIN, D., IMAMURA KAWASAWA, Y., DELGADO-DIAZ, M. R., DE, S., FREIRE, R., SMITS, V. A. & MOLDOVAN, G. L. 2016. HUWE1 interacts with PCNA to alleviate replication stress. *EMBO Rep*, 17, 874-86.
- CROSETTO, N., BIENKO, M., HIBBERT, R. G., PERICA, T., AMBROGIO, C., KENSCH, T., HOFMANN, K., SIXMA, T. K. & DIKIC, I. 2008. Human Wrnip1 is localized in replication factories in a ubiquitin-binding zinc finger-dependent manner. *J Biol Chem*, 283, 35173-85.
- KANU, N., ZHANG, T., BURRELL, R. A., CHAKRABORTY, A., CRONSHAW, J., DACOSTA, C., GRÖNROOS, E., PEMBERTON, H. N., ANDERTON, E. & GONZALEZ, L. 2016. RAD18, WRNIP1 and ATMIN promote ATM signalling in response to replication stress. *Oncogene*, 35, 4009-4019.
- LEUZZI, G., MARABITTI, V., PICHIERRI, P. & FRANCHITTO, A. 2016. WRNIP 1 protects stalled forks from degradation and promotes fork restart after replication stress. *The EMBO journal*, 35, 1437-1451.
- SALIFOU, K., BURNARD, C., BASAVARAJIAH, P., GRASSO, G., HELSMOORTEL, M., MAC, V., DEPIERRE, D., FRANCKHAUSER, C., BEYNE, E., CONTRERAS, X., DEJARDIN, J., ROUQUIER, S., CUVIER, O. & KIERNAN, R. 2021. Chromatin-associated MRN complex protects highly transcribing genes from genomic instability. *Sci Adv*, 7, eabb2947.

SHARMA, S., ANAND, R., ZHANG, X., FRANZIA, S., MICHELINI, F., GALBIATI, A., WILLIAMS, H., RONATO, D. A., MASSON, J. Y., ROTHENBERG, E., CEJKA, P. & D'ADDA DI FAGAGNA, F. 2021. MRE11-RAD50-NBS1 Complex Is Sufficient to Promote Transcription by RNA Polymerase II at Double-Strand Breaks by Melting DNA Ends. *Cell Rep*, 34, 108565.

REVIEWERS' COMMENTS

Reviewer #1 (Remarks to the Author):

The authors have adequately addressed my previous concerns and I do think the authors provide sufficient evidence to support their model and major conclusions of the manuscript.

Reviewer #2 (Remarks to the Author):

The authors have satisfactorily addressed the concerns raised during revision.

Reviewer #3 (Remarks to the Author):

The resubmission of the manuscript titled 'RNAPII-dependent ATM signalling at collisions with replication forks' by Elias Einig et. al., is vastly improved version. I have a few minor issues that need correcting or changing.

1. There is a typo in the sentence: 'Likewise, immunoprecipitation of PCNA and CDC45 yielded increased levels 'or' RNAPII in HUWE1-CS cells compared to HUWE1-WT cells (Supplementary Fig. 1H,I). '
2. The graphs in Supplementary Figure 1H and 1I are incorrectly labelled. If I'm not mistaken the y-axis should both read RNAPII ratio to WT.
3. When referencing Supplementary Figure 2J the authors refer to HUWE1 association with MCM2 whilst the graph reads HUWE1:MCM2
4. The titles for Figures 1, 2 and Supplementary Figures 1 (and the titles for the accompanying sections) do not reflect the data presented.
 - a. Both Figure 1 and Supplementary Figures 1 propose HUWE1 'controls' transcription and DNA replication. These are unsubstantiated claims! I suggest changing the titles to reflect the data more accurately such as 'HUWE1 ubiquitin ligase activity shapes RNA Polymerase II-dependent transcription and prevents TRC'

b. The title of Figure 2 claims HUWE1 and WRNIP1 co-regulate DNA replication and TRCs. How have the authors shown any role for HUWE1 and WRNIP1 in DNA replication? Again, I suggest deleting any reference to regulating DNA replication.

5. On page 4 the authors suggest the loss of both HUWE1 and WRNIP1 cause forks to become unstable. Again, this is unsubstantiated. Here would have been a good opportunity examined fork stability using the fibre assay.

6. I find figure 3E and the description in the manuscript text confusing. I suggest removed data for PARD6B or PTPN1, enlarging the text/figure and changing the labelling of the x-axis.

7. Although not critical to the main conclusion of the paper, I feel the authors should modify their description and interpretation of the correlation between DSB-capture-sequencing and RNAPII distribution in line with the comments provided in the rebuttal.

8. In the discussion the authors write: 'Inactivation of both proteins eliminates TRCs, arguing that in this case replication forks cannot be stabilized at collisions, which leads to impaired ATM signaling and DNA breakage' I find this sentence contradictory! How can the loss of both HUWE1 and WRNIP1 'eliminate' TRCs but the forks are stressed because of collisions? This sentence needs amending.

REVIEWERS' COMMENTS

Reviewer #1 (Remarks to the Author):

The authors have adequately addressed my previous concerns and I do think the authors provide sufficient evidence to support their model and major conclusions of the manuscript.

We thank the reviewer for the comments.

Reviewer #2 (Remarks to the Author):

The authors have satisfactorily addressed the concerns raised during revision.

We thank the reviewer for the comments.

Reviewer #3 (Remarks to the Author):

The resubmission of the manuscript titled 'RNAPII-dependent ATM signalling at collisions with replication forks' by Elias Einig et. al., is vastly improved version. I have a few minor issues that need correcting or changing.

We thank the reviewer for the constructive comments and suggestions.

1. There is a typo in the sentence: 'Likewise, immunoprecipitation of PCNA and CDC45 yielded increased levels 'or' RNAPII in HUWE1-CS cells compared to HUWE1-WT cells (Supplementary Fig. 1H,I). '

We corrected the typo.

2. The graphs in Supplementary Figure 1H and 1I are incorrectly labelled. If I'm not mistaken the y-axis should both read RNAPII ratio to WT.

We corrected the labeling on the graphs in Supplementary Figures 1h and 1i.

3. When referencing Supplementary Figure 2J the authors refer to HUWE1 association with MCM2 whilst the graph reads HUWE1:MCM2

We adjusted the description of this experiment in the main text on page 4.

4. The titles for Figures 1, 2 and Supplementary Figures 1 (and the titles for the accompanying sections) do not reflect the data presented.

a. Both Figure 1 and Supplementary Figures 1 propose HUWE1 'controls' transcription and DNA replication. These are unsubstantiated claims! I suggest changing the titles to reflect the data more accurately such as 'HUWE1 ubiquitin ligase activity shapes RNA Polymerase II-dependent transcription and prevents TRC'

We changed the title of Figure 1.

b. The title of Figure 2 claims HUWE1 and WRNIP1 co-regulate DNA replication and TRCs. How have the authors shown any role for HUWE1 and WRNIP1 in DNA replication? Again, I suggest deleting any reference to regulating DNA replication.

We changed the title of Figure 2 and the title of the corresponding section in the Results.

5. On page 4 the authors suggest the loss of both HUWE1 and WRNIP1 cause forks to become unstable. Again, this is unsubstantiated. Here would have been a good opportunity examined fork stability using the fibre assay.

We agree that our experiments do not directly assess replication fork stability. We attempted to perform the fiber assay as described previously (Balasubramanian et al., 2022) twice, but this experiment failed. Therefore, we rephrased the sentence on page 4.

6. I find figure 3E and the description in the manuscript text confusing. I suggest removed data for PARD6B or PTPN1, enlarging the text/figure and changing the labelling of the x-axis.

We adjusted the description of Figure 3E on pages 4 and 5 for simplicity and increased the font size on the X-axis of the graph for better readability. We chose to remove the data for MSL2 as these values were significantly lower than for the other genes and not clearly visible at the scale.

7. Although not critical to the main conclusion of the paper, I feel the authors should modify their description and interpretation of the correlation between DSB-capture-sequencing and RNAPII distribution in line with the comments provided in the rebuttal.

We adjusted the description of these experiments on pages 5 and 6 of the Results section and added a paragraph on page 9 in the Discussion according to the reviewer's suggestion.

8. In the discussion the authors write: 'Inactivation of both proteins eliminates TRCs, arguing that in this case replication forks cannot be stabilized at collisions, which leads to impaired ATM signaling and DNA breakage' I find this sentence contradictory! How can the loss of both HUWE1 and WRNIP1 'eliminate' TRCs but the forks are stressed because of collisions? This sentence needs amending.

The increase in the PCNA-RNAPII proximity pairs in HUWE1-mutant cells relative to HUWE1-WT cells indicates that some replication forks arrest as a result of TRCs. Additional depletion of WRNIP1 in HUWE1-mutant cells rescues this increase in TRCs, suggesting that in the double mutant cells replication forks do not arrest in the vicinity of RNAPII complexes. We adjusted the phrasing on page 9 accordingly.

References

BALASUBRAMANIAN, S., ANDREANI, M., ANDRADE, J. G., SAHA, T., SUNDARAVINAYAGAM, D., GARZÓN, J., ZHANG, W., POPP, O., HIRAGA, S.-I. & RAHJOUEI, A. 2022. Protection of nascent DNA at stalled replication forks is mediated by phosphorylation of RIF1 intrinsically disordered region. *Elife*, 11, e75047.